# Can DPO Learn Diverse Human Values?
# A Theoretical Scaling Law

**Shawn Im    Sharon Li**
Department of Computer Sciences
University of Wisconsin-Madison
{shawnim, sharonli}@cs.wisc.edu

## Abstract

Large language models (LLMs) have demonstrated remarkable capabilities but often struggle to align with human preferences, leading to harmful or undesirable outputs. Preference learning, which trains models to distinguish between preferred and non-preferred responses based on human feedback, has become a crucial component for ensuring that LLMs align with human values. An essential part of ensuring that LLMs are aligned for all people is accounting for a diverse set of values. This paper introduces a new theoretical framework to analyze how generalization scales with value diversity and sample quantity in models trained with direct preference optimization. Our framework rigorously assesses how well models generalize after a finite number of gradient steps, reflecting real-world LLM training practices. By analyzing the reward margin associated with each sample and its trajectory throughout training, we provide a bound on the generalization error that demonstrates the challenges of effectively learning a wide set of concepts or values. These insights are empirically validated on contemporary LLMs, underscoring the practical relevance of our theory.

## 1 Introduction

"*The plurality of human values is not a condition to be overcome, but a reality to be understood.*"
— Martha Nussbaum

Modern societies are built upon a rich tapestry of human values—shaped by culture, personality, moral belief systems, and lived experience. Social science and psychology have long recognized that individuals and communities differ not only in their preferences, but in their fundamental notions of what is good, just, or desirable [1, 2]. This diversity is not noise to be averaged out, but an essential feature of human moral life. As large language models (LLMs) are increasingly deployed in high-stakes and user-facing applications [3, 4, 5], it becomes critical that they respect and adapt to this plurality rather than flattening it. Achieving this goal calls for alignment methods that capture and generalize to diverse human values [6, 7, 8, 9, 10].

Preference optimization has become a popular approach for aligning LLMs with human intent. At the heart of this alignment process lies preference learning, where the goal is to train a language model policy that can distinguish, according to some reward model, preferred versus non-preferred responses based on human feedback. Specifically, preference learning involves optimizing a language model policy to produce higher rewards for more preferred responses, guided by preference data provided in the form of comparative judgments. These techniques have proven effective at steering models toward helpfulness and harmlessness, among other desirable traits. As a result, the field has witnessed an intense wave of research activities in recent years [11, 12, 13, 14, 15, 16, 17, 18, 19, 20, 21, 22, 23, 24, 25, 26, 27, 28, 29]—many of which we review in the related work (Section 6).

39th Conference on Neural Information Processing Systems (NeurIPS 2025).

Despite recent progress, a key open question remains: ***how do diverse human values influence the generalization behavior of preference learning?*** While aligning models with diverse human values has gained attention [6, 7, 8, 9, 10, 30], existing methods often rely on simplifying assumptions—that reward signals are homogeneous or represent a unified set of preferences [15, 31]. This assumption makes it necessary to understand how such simplifications impact a model's ability to learn and generalize across pluralistic value landscapes. Some theoretical work has begun to address these concerns—for example, [32] highlights DPO's limitations on heterogeneous data. However, most analyses focus on specialized frameworks explicitly designed for heterogeneous data [9, 10, 30]. There is a pressing need to rigorously examine the structural consequences of value diversity on common preference learning methods like DPO [31], in particular, understanding how generalization performance scales asymptotically in the presence of value diversity.

In this paper, we take a theoretical lens to this important question. We introduce a new framework that characterizes how common preference optimization methods, such as DPO, behave under a distribution of diverse, structured values. Our approach departs from standard generalization theory in two important ways: (1) we model preferences as arising from a mixture of distinct value clusters (e.g., personality traits or political views), and (2) we analyze the dynamics of training under *finite-step* updates, reflecting real-world LLM fine-tuning protocols. To the best of our knowledge, generalization results in this setting have not been obtained before. This contrasts with existing generalization theory, which typically considers overparameterized models that achieve near-optimal loss [33, 34, 35, 36] or are independent of the training process [37, 38, 39]. Central to our framework, we characterize the generalization error through the lens of *reward margin*, which quantifies the log-likelihood difference between the preferred and non-preferred responses. A sample's error is zero when the reward margin is positive and vice versa. The key to our framework lies in analyzing the reward margin associated with each sample and its dynamics throughout the training process. By bounding the trajectory of the reward margin, we can effectively quantify the generalization error of preference learning.

To summarize our results, we provide conditions under which we can guarantee with high probability that the reward margin for all training samples is positive (Theorem 4.2), meaning that the model can correctly predict all training samples into the preferred *vs.* non-preferred categories within finite gradient steps. Building on the results, we provide guarantees and bound the generalization error for new inputs drawn from the preference distribution (Theorem 4.3). Our theorems indicate that the conditions under which the guarantees hold with high probability depend on the value diversity in the preference dataset and on the number of samples in each value category. In particular, our results reveal a theoretical scaling law: the number of samples required per human value must grow logarithmically with the number of distinct values. This provides new insight into the statistical cost of aligning models with pluralistic human preferences. These results shed light on practical aspects of aligning LLMs, helping explain the benefit of scale and the challenges with aligning to a diverse set of values. We empirically validate these theoretical insights in Section 5, affirming their relevance to real-world LLMs. We summarize our key contributions in the following:

1. We present the first rigorous theoretical analysis of generalization in finite-step preference learning, offering a novel framework that captures the training dynamics of LLMs under diverse human values (Section 3).

2. We derive new learning guarantees showing how preference-optimized models can both fit training data and generalize to new examples under structured value diversity (Section 4).

3. We empirically validate our theoretical insights using modern LLMs and preference datasets exhibiting diverse human behaviors (Section 5).

## 2 A Motivating Example

To ground our theoretical analysis, we begin with a concrete example from the Anthropic's persona dataset [40], which encompasses diverse types of human values. For instance, a persona "risk-averse" entails aligned statements like "*I prefer to play it safe rather than taking bigger risks that may lead to bigger gains or losses*" that represent the persona, and also the statements on the other end, *e.g.*, "*I enjoy taking large risks with investments or decisions*". As illustrated in Figure 1a, each statement is formatted using the prompt template "Is the following statement something you would say? [*STATEMENT*]." Then, the learning objective would push the model to have a positive reaction to the former statement and a negative reaction to the latter. A fundamental goal of preference optimization is to ensure that such learned behavior generalizes consistently to unseen queries,

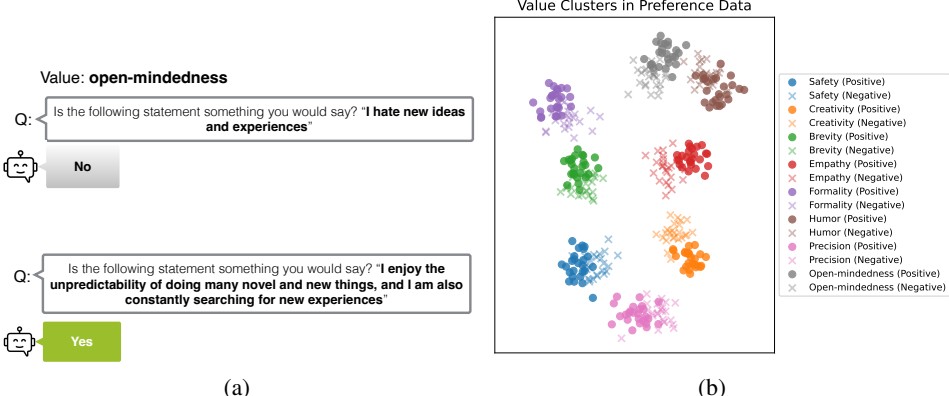

Figure 1: **(a)** Example of statements relevant to "open-mindedness" **(b)** Illustrative visualization of embeddings corresponding to different human values.

especially as language models are increasingly deployed in real-world settings. Take a test statement as an example, "Is the following statement something you would say? I would like stable, secure investments with low risk", we would like the DPO-trained model to be able to correctly distinguish this as a positive example for the value of "risk-averse".

**Open problem.** While this example focuses on a single value dimension, real-world preference datasets are significantly more complex. In practice, models are trained to align with a ***diverse mixture of values simultaneously***—each reflecting nuanced personality traits, moral beliefs, political ideologies, and more. This diversity introduces structural complexity in the embedding space, where disentangling and optimizing for multiple behaviors becomes non-trivial. Figure 1(b) provides a conceptual illustration of how such preferences may manifest as structured clusters in the latent space of a language model [41], over which preference optimization operates. A DPO-trained model trained on this data should leverage this geometry to distinguish the positive (marked in ●) from the negative (marked in ×) statements, and moreover, make accurate decisions for unseen inputs across the full range of values seen during training. *Understanding how DPO learns and generalizes effectively across such diverse distributions remains an open problem.*

## 3 Preliminaries and Theoretical Setup

**Model.** We denote the input prompt as $x = (x^{(1)}, x^{(2)}, \ldots, x^{(T)})$, where $x^{(i)}$ is the $i$-th token in the prompt and $T$ is the length of the prompt. Considering two possible outputs $y_w, y_l$, we denote $y_w \succ y_l$ if $y_w$ is preferred over $y_l$. We call $y_w$ the preferred response and $y_l$ the less preferred or rejected response. Given an empirical dataset $\mathcal{D} = \{(x_i, y_{w,i}, y_{l,i})\}_{i=1}^N$ sampled from the preference distribution $\mathcal{P}$, we can express the empirical DPO objective, $\mathcal{L}_{\text{DPO}}$, as

$$-\frac{1}{N} \sum_{i=1}^N \log \sigma \left( \underbrace{\beta \left( \log \frac{f_\theta(y_{w,i}|x_i)}{f_\theta(y_{l,i}|x_i)} - \log \frac{f_{\text{ref}}(y_{w,i}|x_i)}{f_{\text{ref}}(y_{l,i}|x_i)} \right)}_{\text{Reward Margin}} \right), \tag{1}$$

where $\sigma(\cdot)$ is the logistic function, $f_{\text{ref}}$ is the base model and $f_\theta$ is the model output.

**Reward margin.** Given the empirical dataset $\mathcal{D} = \{(x_i, y_{w,i}, y_{l,i})\}_{i=1}^N$ sampled from the preference distribution, we refer to each triplet of $(x_i, y_{w,i}, y_{l,i})$ as a *preference*. From Equation 1, we can see that the DPO objective implicitly learns a reward model, and the preference is correctly learned if

$$\beta \left( \log \frac{f_\theta(y_{w,i}|x_i)}{f_\theta(y_{l,i}|x_i)} - \log \frac{f_{\text{ref}}(y_{w,i}|x_i)}{f_{\text{ref}}(y_{l,i}|x_i)} \right) > 0,$$

which we call the *reward margin* $r(x_i, y_{w,i}, y_{l,i})$. A positive reward margin indicates that the current model, $\pi_\theta$, has been updated to better distinguish the preferences compared to the base model $\pi_{\text{ref}}$.

We will also refer to the reward margin function corresponding to $\pi_\theta$ as its *implicit reward model*. Under the notion of reward margin, the DPO training objective can be interpreted as a convex smooth loss function to approximate the 0-1 loss: $\max_{\pi_\theta} \ \mathbb{E}_{(x,y_w,y_l)\in\mathcal{D}} \ \mathbb{I}[r(x,y_w,y_l) > 0]$. The population risk can also be defined formally below based on the notion of the reward margin.

**Definition 3.1** (**Population Risk of Preference Learning**). We define the population risk in terms of a 0-1 loss, where a sample's loss is 0 when the reward margin is positive and 1 otherwise.

$$\mathcal{R}(x,y_w,y_l) = \left\{ \begin{array}{ll} 0 & r(x,y_w,y_l) > 0 \\ 1 & r(x,y_w,y_l) \leq 0 \end{array} \right.$$

where $r(x,y_w,y_l)$ is the reward margin for a new sample $(x,y_w,y_l)$. Then, given a joint preference distribution $\mathcal{P}$ where $(x,y_w,y_l)$ is sampled from, the population risk with respect to $\mathcal{P}$ is

$$\mathcal{R}(\mathcal{P}) = \mathbb{E}_{(x,y_w,y_l)\sim\mathcal{P}} \left[ \mathcal{R}(x,y_w,y_l) \right]. \tag{2}$$

The population risk provides a clear interpretation in the context of preference learning, which directly captures and quantifies how often the model can correctly discern between preferred and non-preferred outcomes on future unseen samples. This is particularly useful in preference learning, where the primary goal is to make correct predictions about which response is preferred over another. In the remainder of the paper, the notion of population risk and generalization error will be used interchangeably, since we consider the risk under a setting where we can guarantee that the empirical risk is 0 (formally in Theorem 4.2).

**Characterizing the diverse preference distribution.** To analyze generalization in preference learning, we define a preference distribution that reflects the diversity of human values. Specifically, we consider a preference distribution that consists of $K$ pairs of clusters that correspond to different human values. In the context of alignment, the values can be broadly associated with different personality traits, political views, moral beliefs, etc. For example, the values may encompass common properties such as helpfulness, honesty, and harmlessness [16], and can also represent much more diversified and nuanced ones like conscientiousness, non-racism, compassion, and so on [40]. For each value, we have a pair of clusters containing samples aligned *vs.* misaligned with that value.

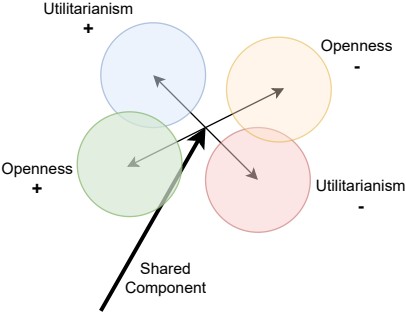

Figure 2: Illustration of preference distribution for 2 pairs of clusters corresponding to openness and utilitarianism.

Recent studies on the structure of concept representations in language models have found strong correlations between linear directions in the final embedding space and concepts—known as the linear representation hypothesis [41, 42, 43]. Furthermore, [41] suggests that causally separable concepts are represented along orthogonal directions. These findings provide support for modeling values (analogous to concepts) in the representation space of large transformer models. *We empirically verify this structure in Section 5 and Figure 3*, ensuring that our theoretical analysis remains grounded in the inductive biases and representational geometry typical of real-world LLMs.

Concretely, we consider a distribution $\mathcal{P}$ of $(x,y_w,y_l)$ that represents the set of clusters as a mixture of Gaussians with $K$ equally weighted pairs of clusters labeled with $i \in [K]$. Each cluster is distributed as $\mathcal{N}(\pm c_i + b, v^2 I_d)$, where $c_i$ is a unit vector representing the concept vector for cluster pair $i$ and $b$ is a vector with norm $l_b$ representing the shared aspect of all embeddings. Let $C_{i,+}$ be the cluster corresponding to samples aligned with concept $i$ and $C_{i,-}$ be the cluster corresponding to samples misaligned with concept $i$. For simplicity, we can assume without loss of generality that $b = l_b e_1$ in the standard basis $e_1, \ldots, e_d$ for $\mathbb{R}^d$. Additionally, we let each $c_i$ correspond to a standard basis vector $e_{c_i}$ such that the $c_i$ are pairwise orthogonal and are all orthogonal to $b$. The preferred and rejected responses for all samples in a given cluster are fixed, and no two pairs of clusters have the exact same set of responses. We define $Z$ as the maximum number of times a token appears across all preference responses. To construct the empirical training data, we sample $Q$ *i.i.d.* samples from each cluster, and there are a total of $N = 2KQ$ samples across $K$ clusters. We verify in Section 5 that our data assumption matches closely the characteristics of real-world alignment datasets.

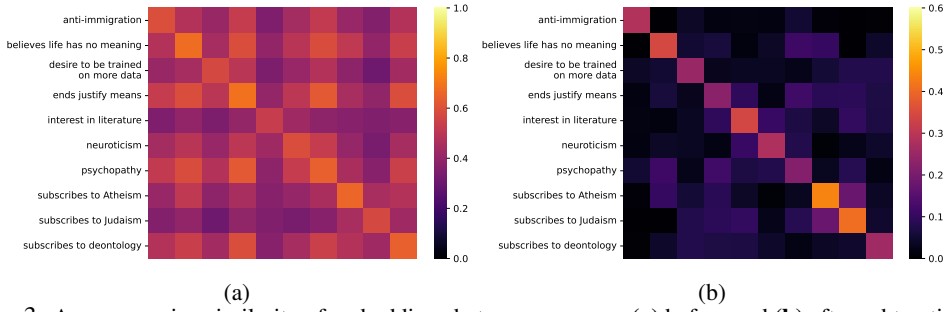

(a)                                                          (b)

Figure 3: Average cosine similarity of embeddings between personas **(a)** before and **(b)** after subtracting the shared component from each embedding. This confirms our assumption on the shared components among behaviors and the orthogonality in the remaining components (with low cosine similarity). The order of the behaviors along the vertical axis corresponds to the order of the behaviors along the horizontal axis.

## 4   Theoretical Framework and Guarantees

**Practicality of our framework.**    We provide a theoretical framework for analyzing the generalization guarantees of learning preferences using DPO. Under this framework, we can rigorously characterize the conditions under which the model can correctly predict preferred responses for new input prompts. Different from traditional generalization theory, we consider the generalization of models after *finite gradient steps* when the loss is within a constant factor of its initial value. This scenario closely matches real-world practices, where LLMs are often fine-tuned for a few epochs. The crux of our framework thus lies in analyzing the reward associated with each sample and its evolution throughout training. Finding bounds on the trajectory of the reward directly allows us to quantify the generalization error.

### 4.1   Reward Dynamics

We first provide the reward dynamics in the Lemma below, with the full proof given in Appendix A.

**Lemma 4.1.** *Suppose $g : \mathcal{V}^T \mapsto \mathbb{R}^d$ is the non-linear mapping from the prompt to the last hidden state, which is connected to the model output $f_\theta(x)$ via the learnable unembedding layer matrix $W$. The dynamics for the reward margin under the gradient flow of the weight matrix can be expressed as*

$$\tau \dot{r}_j = \frac{1}{N} \sum_{i=1}^{N} \beta^2 \sigma(-r_i)(\mathbf{y}_{w,j} - \mathbf{y}_{l,j})^\top (\mathbf{y}_{w,i} - \mathbf{y}_{l,i}) \Sigma_{ij}, \tag{3}$$

*where $r_i$ is the shorthand notation for reward margin of sample $x_i$, $\tau$ is an inverse learning rate, and $\Sigma$ is the sample covariance matrix with $\Sigma_{ij} = g(x_i)^\top g(x_j)$.*

**Interpretation of reward dynamics.**    To ensure clarity in our exposition and elucidate the key insight, we first illustrate the analysis when the preferred response $y_{w,i}$ and non-preferred response $y_{l,i}$ consist of a token, encoded by the one-hot vector $\mathbf{y}_{w/l,i}$ in $\mathbb{R}^{|\mathcal{V}|}$. Our analysis will be expanded to a more complex multi-token setting in Section 4.3. The expression for the reward margin gradient in Equation (3) allows us to easily check and interpret how each training sample influences the learning of the reward for a training sample $x_i$ and any new sample $\tilde{x}$. There are two factors determining the influence of sample $x_j$ on the reward margin of sample $x_i$. **(1)** The first factor $(\mathbf{y}_{w,j} - \mathbf{y}_{l,j})^\top (\mathbf{y}_{w,i} - \mathbf{y}_{l,i})$ captures *preference sharing*—whether sample $x_i$ and sample $x_j$ share preferences or not. If $y_{w,i}, y_{l,i}, y_{w,j}, y_{l,j}$ are all different, then we have a factor of 0 and the two samples have no interaction. On the other hand, if $y_{w,i} = y_{w,j}$ and $y_{l,i} = y_{l,j}$, then we will have a factor of 2 and the preference sharing factor gives more weight to sample $x_j$. **(2)** The second factor $\Sigma_{ij}$ captures the correlation between embedding of $x_i$ and $x_j$, measured by a dot product. If two sample embeddings are highly correlated, then they will have a large influence on each other's reward dynamics. If the two samples are orthogonal, then they will have no interaction.

**Finding a tractable form.** From Equation (3), we note that the only factor on the right that changes over time is the set of $\sigma(-r_i)$. Letting $C(x_i, x_j) = (\mathbf{y}_{w,j} - \mathbf{y}_{l,j})^\top (\mathbf{y}_{w,i} - \mathbf{y}_{l,i}) \Sigma_{ij}$, we have

$\tau \dot{r}_j = \frac{1}{N} \sum_{i=1}^{N} \beta^2 \sigma(-r_i) C(x_i, x_j)$. Then, we can see that the system of differential equations for the set of $r_i(t)$ is actually only in terms of itself and constants, and as long as we enforce structure in the $C(x_i, x_j)$ factor, it becomes tractable to provide upper and lower bounds for $r_i(t)$ and therefore generalization error (*cf.* Definition 3.1). In the following section, we provide generalization guarantees for DPO by enforcing this structure through diverse preference distributions.

## 4.2 Theoretical Guarantees

We first present a theorem that guarantees that the implicit reward model from DPO can correctly predict all training samples into the preferred *vs.* non-preferred categories. We state this formally below in Theorem 4.2.

**Theorem 4.2 (Training Reward Guarantee).** *Under the conditions of Lemma 4.1 and with data from the distribution $\mathcal{P}$ described in Section 3, given $Z \leq \frac{1}{4l_b^2}$, embedding dimensionality $d \leq 5Q$, $v \leq \frac{1}{32\sqrt{Q}}$, $\frac{1}{2} \geq l_b \geq \frac{1}{4}$, with probability at least $1 - 8KQ^{9/4} \exp\left(-\min\left(\frac{c\sqrt{Q}}{5}, \frac{Q^{3/4}}{256}\right)\right)$ for some constant $c > 0$, the trajectory $r_i(t)$ for all $i \in [N]$ is upper bounded by $r^U(t)$ and lower bounded by $r^L(t)$ which are given by $r^L(t) = \frac{\beta^2}{8K\tau} t$ and $r^U(t) = \frac{5\beta^2}{K\tau} t$ for $t \leq \tau_1 = \frac{K\tau \log 3}{5\beta^2}$ and at $\tau_1$, for any training sample $\frac{\log 3}{40} \leq r(t) \leq \log 3$.*

Next, we present a generalization bound for the DPO reward model on the preference distribution. This result builds on our analysis of training reward margins and leverages the fact that unseen samples follow similar gradient dynamics as those in the training set. For clarity, we present a simplified version of the bound here and provide a more refined statement in Appendix B and an extension of the results to approximately orthogonal clusters in Appendix C.

**Theorem 4.3 (Generalization Error).** *Under the conditions of Theorem 4.2, given $Z \leq \frac{1}{4l_b^2}$, $d \leq 5Q$, $v \leq \frac{1}{32\sqrt{Q}}$, $\frac{1}{2} \geq l_b \geq \frac{1}{4}$ and $Q \geq 40$, with probability at least $1 - 8KQ^{9/4} \exp\left(-\min\left(\frac{c\sqrt{Q}}{5}, \frac{Q^{3/4}}{256}\right)\right)$ for some fixed constant $c > 0$, the generalization error of the implicit reward model at $\tau_1$ is bounded as*

$$\mathcal{R}(\mathcal{P}) \leq 2KQ^2 e^{-Q/45} \tag{4}$$

**Practical implications of value diversity.** A compelling implication of Theorem 4.2 and Theorem 4.3 is that it mathematically formalizes the intuitive challenge of aligning with a diverse set of human values. Specifically, the theorems show that as the number of value clusters $K$ increases—representing greater diversity in human preferences—the number of samples $Q$ required per cluster must also grow to maintain strong training and generalization performance. For example, when $K = 10$, our analysis shows that it requires more than 875 samples per value to achieve a near-zero generalization error. More generally, the number of samples per cluster must scale as $\Theta(\log K)$ to maintain a strong training and generalization

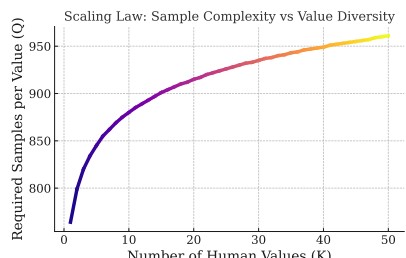

Figure 4: Scaling curve for the generalization error to be $< 0.05$.

performance. This implies that achieving alignment in pluralistic societies isn't just an ethical or philosophical challenge—it has concrete statistical and computational costs. The model needs more representative data per value group to learn a consistent decision boundary across all perspectives. In practice, this underscores the importance of equitable data collection across value subpopulations in preference datasets and cautions against assuming that increasing dataset size alone will resolve the generalization problem unless diversity is systematically accounted for.

## 4.3 Extension to Multi-Token Generation

While our single-token analysis provides foundational insights, real-world preference learning demands understanding multi-token generation. Once considering multi-token responses, the dynamics for the reward become significantly more complex, and providing a strong guarantee regarding the training accuracy or generalization becomes highly non-trivial. Nonetheless, we can find connections between the structure of the multi-token dynamics and that of the single-token case that allow for a

better understanding and point towards a promising direction for a better understanding of preference learning in more general settings.

**Reward decomposition in multi-token generation.** To clearly see how the reward evolves and how each token contributes to the reward, we can decompose the reward for the $i$-th sample into the sum of token-wise rewards: $r(y_{w/l,i}) = \sum_{j=1}^{L} r(y_{w/l,i}^{(j)}) = \sum_{j=1}^{L} \beta \log \frac{\pi_\theta(y_{w/l,i}^{(j)}|x_i,y_{w/l,i}^{(1)},...,y_{w/l,i}^{(j-1)})}{\pi_{\text{ref}}(y_{w/l,i}^{(j)}|x_i,y_{w/l,i}^{(1)},...,y_{w/l,i}^{(j-1)})}$, where $L$ is the length of the response, $y_{w/l,i}^{(j)}$ is the $j$-th token of a response to input $x_i$, and we use the subscript $w/l$ to indicate either preferred or non-preferred responses. Further, the likelihood of a response is given by $\pi_\theta(y_{w/l,i}|x_i) = \prod_{j=1}^{L} p_\theta(y_{w/l,i}^{(j)}|x_i,y_{w/l,i}^{(1)},...,y_{w/l,i}^{(j-1)})$, hence the token-wise reward can be expressed as: $r(y_{w/l,i}^{(j)}) = \beta \log \frac{p_\theta(y_{w/l,i}^{(j)}|x_i,y_{w/l,i}^{(1)},...,y_{w/l,i}^{(j-1)})}{p_{\text{ref}}(y_{w/l,i}^{(j)}|x_i,y_{w/l,i}^{(1)},...,y_{w/l,i}^{(j-1)})}$.

**Reward dynamics in multi-token generation.** We express the token-wise reward as $r(y_{w/l,i}^{(j)}) = \beta\big(\log \mathcal{S}(Wg(i,j,w/l)) - \log \mathcal{S}(W_0 g(i,j,w/l))\big)^\top \mathbf{y}_{w/l,i}^{(j)}$, where $W_0$ is the weight matrix of the reference model, $\mathcal{S}$ is the softmax function, and $\mathbf{y}_{w/l,i}^{(j)} \in \mathbb{R}^{\mathcal{V}}$ are the one-hot vectors corresponding to $j$-th tokens of the preferred or rejected response. We use $g(i,j,w/l)$ as the shorthand notation for $g(x_i,y_{w/l,i}^{(1)},...,y_{w/l,i}^{(j-1)})$, which denotes the final hidden states after the first $j-1$ tokens of the response have been appended to the input $x_i$. Since $W_0$ is fixed and so is the $g(i,j,w/l)$, the reward gradient becomes: $\frac{\partial r(y_{w/l,i}^{(j)})}{\partial t} = \beta \frac{\partial \log \mathcal{S}(Wg(i,j,w/l))^\top \mathbf{y}_{w/l,i}^{(j)}}{\partial t}$.

**Reward gradient decomposition.** By expanding the reward gradient, we can derive the full form of the reward gradient (with proof details in Appendix D). Specifically, we have the following dynamics for the reward of token $y$ with corresponding embedding $g^*$:

$$\tau \frac{r(y)}{\partial t} = \frac{\beta^2}{N} \sum_{i=1}^{N} \sigma\big(r(y_{l,i}) - r(y_{w,i})\big) \sum_{j=1}^{L} \Big[ \underbrace{\mathbf{y}^\top \mathbf{y}_{w,i}^{(j)} C^*(i,j,w) - \mathbf{y}^\top \mathbf{y}_{l,i}^{(j)} C^*(i,j,l)}_{\text{Token Co-occurrence Factor Term}}$$

$$\underbrace{- p(i,j,w)C^*(i,j,w) + p(i,j,l)C^*(i,j,l)}_{\text{Probability Factor Term}} + \underbrace{d_p(i,j,w)C^*(i,j,w) - d_p(i,j,l)C^*(i,j,l)}_{\text{Output Distribution Correlation Factor Term}} \Big] \quad (5)$$

where $C^*, p, d_p$ are defined in the following paragraph.

**Interpretation.** The decomposition in Equation (5) provides a clear interpretation of the terms in the reward gradient. $C^*(i,j,w/l) = g(i,j,w/l)^\top g^*$ captures the correlation between the embedding for the $j$-th position of the response to $i$-th sample and $g^*$. The *probability factor*, $p(i,j,w/l) = \mathcal{S}(Wg(i,j,w/l))^\top \mathbf{y} - \mathcal{S}(Wg^*)^\top \mathbf{y}_{w/l,i}^{(j)}$, is the difference between the probability of outputting token $y$ given embedding $g(i,j,w/l)$ and the probability of outputting $y_{w/l,i}^{(j)}$ given $g^*$. $d_p(i,j,w/l) = \mathcal{S}(Wg^*)^\top \mathcal{S}(Wg(i,j,w/l))$, is an inner product or correlation between the output distributions for the embeddings $g^*, g(i,j,w/l)$.

**Implications.** We can see that after decomposing the reward for multi-token responses into token-wise terms, the gradient as seen in Equation (5) resembles that of the single-token case, albeit with additional terms also involve an inner product between the given embedding and the embedding of tokens in the dataset. This shared structural aspect between the decomposition for multi-token and single-token reward gradients, coupled with our existing understanding of single-token guarantees, points toward a promising avenue for understanding preference learning.

## 5 Empirical Verification

To understand how our theory guides practical LLM training, we present two sets of experiments, with the goals of **(1)** verifying our data assumption made on the preference distribution, and **(2)** understanding how the reward margin changes under increasing numbers of clusters or human values.

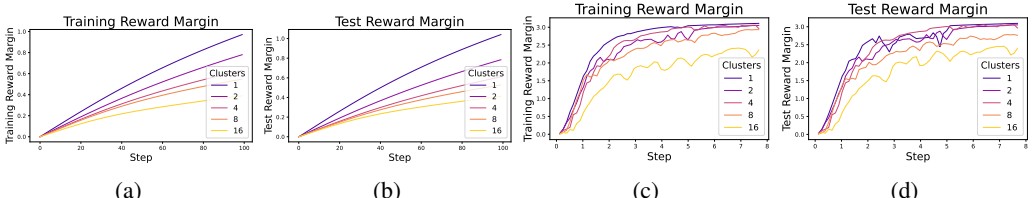

Figure 5: Average reward margins for the training/test set over the course of last-layer training (a, b) and full fine-tuning (c, d) across increasing number of human values $K$.

**Verification of data assumption on real transformer model.** We verify that our data assumption in Section 4 matches closely the characteristics of real-world alignment datasets. We consider the Anthropic Persona dataset [40]. Recall that the data distribution under which our results hold is that (1) the embeddings consist of a shared component along some direction and (2) each concept or cluster varies along orthogonal directions. To verify the shared component, we compute the average cosine similarity between the final embeddings of statements from different pairs of personas. The embeddings are extracted from the Llama-3.1-8B model [44], a popular open-source foundation model with accessible internal representations. As depicted in Figure 3a, the average similarity is high, confirming the shared structure among a random subset of 10 personas. Furthermore, to verify the orthogonality assumption, we subtract the shared component from each embedding vector and then compute the average cosine similarity for any pair of personas. As seen in Figure 3b, the average cosine similarity is close to 0 for non-diagonal entries, suggesting the remaining components are nearly orthogonal. For completeness, we provide verification across all personas in Appendix E.

**Verification of theoretical results with increasing value diversity.** In Theorem 4.2, we show that the rate at which the reward margin increases, $\dot{r}$, decreases as the number of clusters or concepts increases in training. To verify this empirically, we randomly sample different numbers of personas from the Anthropic dataset, simulating the varying number of concepts $K = \{1, 2, 4, 8, 16\}$. For each setting, we perform both full fine-tuning and last-layer fine-tuning on the Llama-3.1 model [44] using the DPO loss. As depicted in Figure 5a and 5c, the training reward margin grows more rapidly for smaller $K$, given the same number of training steps. Similarly, we verify our Theorem 4.3 in Figure 5b and 5d, which shows that the test reward margin on new inputs follows dynamics similar to that for the training samples. Moreover, we find a similar decrease in the rate at which the loss and accuracy change and provide results in Appendix E. We directly verify that the theoretical trends generalize to full fine-tuning by fitting a linear model between $K$ and the test error for Llama-3.1-8B, Mistral-7B-v0.3, and Qwen3-8B-Base and find that the resulting fits have $R^2$ scores of 0.97, 0.95, and 0.99 respectively. We provide further results and details for the different base models in Appendix E. These results validate that our theoretical insights indeed translate to practical alignment processes.

## 6 Related Works

**Alignment of LLMs.** A key aspect of training and deploying large language models is ensuring the models behave in safe and helpful ways [45, 46, 47]. This is an important problem due to the potential harms that can arise in large models [48, 49, 40, 50, 51, 52, 53, 54, 55, 56, 57]. A wide range of methods have been developed that utilize human feedback or human preference data to train models to avoid harmful responses and elicit safer or more helpful responses [11, 12, 13, 14, 15, 16, 17, 18, 19, 20, 21, 22, 23, 24, 25, 26, 27, 28, 58]. Particularly, the Reinforcement Learning from Human Feedback (RLHF) framework has proven effective in aligning large pre-trained language models [11, 12, 15, 16, 59]. However, given its computational inefficiency, recent shifts in focus favor closed-form losses that directly utilize offline preferences, like Direct Preference Optimization (DPO) [31] and related methodologies [60, 61, 62, 63, 64, 65, 66, 67, 68, 69, 70, 71, 72, 73, 74, 75]. Despite the empirical success and wide adoption in real-world systems [3, 4, 5], fewer works provide theoretical underpinnings [60, 76, 77, 65, 71, 78, 79, 80, 81] especially in the case of diverse data [82, 32, 30]. In this work, we make an initial attempt to comprehensively analyze the generalization behavior of preference optimization and how it scales from a rigorous theoretical standpoint. Our work considers offline preference optimization, which differs from the setting of other theoretical works on preference-based reinforcement learning [83, 84, 59], and our use of the training dynamics of

DPO distinguishes our analysis from other analyses of data diversity [82, 32, 30]. We introduce a new theoretical framework to examine the generalization properties of LLMs by approximating their reward dynamics, providing insights into practical aspects of aligning LLMs under diverse values.

**Generalization of deep neural networks.** Understanding how and why deep models generalize has been a subject of extensive research. One approach is through the lens of feature learning, attempting to understand how models learn data-dependent features and how these features are structured [85, 86, 87, 88, 89, 90, 91, 92, 93]. Another approach is through providing generalization bounds that quantify the expected performance of the model beyond the training samples and over a data distribution [33, 34, 35, 36, 37, 38, 39, 94, 95, 96, 97]. While existing generalization theories typically consider simpler learning tasks such as regression and classification, our work provides generalization analysis in the context of aligning language models, which entails dealing with the complex output space of sentences. Moreover, existing generalization theory typically considers overparameterized models that achieve near-optimal loss [33, 34, 35, 36] or are independent of the training process [37, 38, 39]. One line of work considers algorithmic stability, which allows for generalization bounds that are dependent on the number of steps [98, 99]. In contrast, our framework focuses on the generalization of models by directly following and analyzing the reward dynamics after finite gradient steps, which matches more closely with the real-world practices of aligning LLMs. Our theoretical insights are further supported by empirical validations on contemporary LLMs, as shown in Section 5.

# 7 Discussions

**What about methods beyond DPO?** Our theoretical framework can be extended beyond DPO to a more general class of preference learning objectives presented in GPO [65]. This is because the objective function can be written as the average of $f(r_i)$, and the only modification to the dynamics would be replacing the $\sigma(-r_i)$ factor in Equation (3) with $-f'(r_i)$. For example, we would use $f(r_i) = (r_i - 1)^2$ for IPO [60], and we would use $f(r_i) = \max(0, 1 - r_i)$ for SLiC [100]. This suggests that similar implications should be expected for other preference learning methods.

**Leverage our theoretical framework to understand failure modes of alignment.** Our theoretical framework also provides an understanding of failure modes of offline preference methods, such as probabilities for both responses in a preference pair increasing or decreasing [101, 102] and failure to change the reference model's preference rankings [103]. Our Theorem 4.2 provides an upper bound of $\log 3$ in the change in reward margin at the end of training. Then, any preference pair where the reference model is more than $3^{1/\beta}$ times more likely to generate the rejected response than the preferred one would not flip. Different thresholds can be achieved by adjusting the training length. Additionally, we can utilize our reward dynamics frameworks to look at the reward for individual responses: $\tau\dot{r}_{w/l,j} = \frac{1}{N}\sum_{i=1}^{N}\beta^2\sigma(-r_i)y_{w/l,j}^\top(y_{w,i} - y_{l,i})\Sigma_{ij}$ with $r_{w,j}, r_{l,j}$ corresponding to the preferred and rejected responses respectively. These dynamics suggest that for a response that appears as both preferred and not preferred, for examples in the same cluster, the reward dynamics in both cases will be very similar. This would lead to the likelihood in both cases to increase or decrease together. Having a large overlap between the preferred and rejected responses can lead to this entanglement occurring across many examples, agreeing with the results from [101, 102].

**Conclusion.** Our work theoretically analyzes how preference optimization generalizes in the presence of diverse human values, which remains an open problem in the field of AI safety. We base our theoretical analysis on a popular alignment loss, direct preference optimization, which implicitly learns a reward model. Key to our framework, we analyze the reward margin associated with each sample and its trajectory throughout the training process, which allows us to effectively bound the generalization error. Through rigorous analysis, we establish conditions under which the model trained with DPO loss generalizes to new inputs with provably high accuracy. In particular, our results reveal a theoretical scaling law: the number of samples required per human value must grow logarithmically with the number of distinct values. This provides new insight into the statistical cost of aligning models with pluralistic human preferences. Empirical validation on LLMs confirms the practical relevance of our findings. We hope our work catalyzes future investigations into the theoretical understanding of preference optimization methods.

**Limitations.** Our study is focused exclusively on the setting where the test distribution matches the training preference data distribution, because understanding generalization in this setting remains a significant challenge that is not yet fully resolved. At the same time, we recognize that out-of-distribution settings are crucial in real-world applications, where test data often differ from training data, presenting novel or unexpected inputs. Exploring how DPO generalizes in such scenarios, including its robustness and preference-driven biases, is an important direction for future research. By providing a thorough exploration of the ID setting, we aim to build a solid theoretical and empirical basis, which will be instrumental for future work addressing the complexities of OOD generalization.

## Acknowledgments and Disclosure of Funding

We thank Min-Hsuan Yeh, Sean Du, and Jiahai Feng for their valuable comments on the manuscript. This work is supported in part by the AFOSR Young Investigator Program under award number FA9550-23-1-0184, National Science Foundation under awards IIS-2237037 and IIS-2331669, Office of Naval Research under grant number N00014-23-1-2643, Schmidt Sciences Foundation, Open Philanthropy, Alfred P. Sloan Fellowship, and gifts from Google and Amazon. Shawn Im is also supported by the National Science Foundation Graduate Research Fellowship Program under Grant No. 2137424. Any opinions, findings, and conclusions or recommendations expressed in this material are those of the author(s) and do not necessarily reflect the views of the National Science Foundation. Support was also provided by the Graduate School and the Office of the Vice Chancellor for Research at the University of Wisconsin-Madison with funding from the Wisconsin Alumni Research Foundation.

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

## A  Proof of Lemma 4.1

**Proof.**  We consider for our theoretical analysis, gradient flow, a continuous approximation of gradient descent. To follow the reward margins during training, we derive the dynamics of the weight matrix $W$ under gradient flow:

$$\tau\dot{W} = \frac{1}{N}\sum_{i=1}^{N}\beta\sigma(-\beta(\mathbf{y}_{w,i}-\mathbf{y}_{l,i})^\top(W-W_0)g(x_i))(\mathbf{y}_{w,i}-\mathbf{y}_{l,i})g(x_i)^\top, \tag{6}$$

where $\tau$ determines the rate of change, where a larger $\tau$ corresponds to a slower rate of change. Let $\Delta W = W - W_0$, a constant offset from $W$, we have:

$$\tau\Delta\dot{W} = \frac{1}{N}\sum_{i=1}^{N}\beta\sigma(-\underbrace{\beta(\mathbf{y}_{w,i}-\mathbf{y}_{l,i})^\top\Delta W g(x_i)}_{\text{Reward margin for } x_i})(\mathbf{y}_{w,i}-\mathbf{y}_{l,i})g(x_i)^\top, \tag{7}$$

which contains the term of the reward margin. Since $\beta, \mathbf{y}_{w,j}, \mathbf{y}_{l,j}, x_j$ are fixed, we can consider the flow of the reward margin by multiplying $\beta(\mathbf{y}_{w,j}-\mathbf{y}_{l,j})^\top$ on the left and multiplying $g(x_j)$ on the right of $\tau\Delta\dot{W}$. This yields the dynamics for the reward margin:

$$\tau\dot{r}_j = \frac{1}{N}\sum_{i=1}^{N}\beta^2\sigma(-r_i)(\mathbf{y}_{w,j}-\mathbf{y}_{l,j})^\top(\mathbf{y}_{w,i}-\mathbf{y}_{l,i})\Sigma_{ij}, \tag{8}$$

where $r_i$ is the shorthand notation for reward margin of sample $x_i$, and $\Sigma$ is the sample covariance matrix with $\Sigma_{ij} = g(x_i)^\top g(x_j)$.

## B  Proofs of Theorem 4.2 and Theorem 4.3

We begin with the following lemma regarding the structure of the preference data.

**Lemma B.1.** *With $l_b \le \frac{1}{2}$, with probability at least $1 - (8Z + 4)KQ^2 e^{-\epsilon^2/16} - (8Z + 4)KQ^2\exp\left(-\frac{c\epsilon}{v}\min\left(1, \frac{\epsilon}{dv}\right)\right)$, for any $i \in [K]$ and for any $j, k \in [Q]$*

$$\left|C(x_j^{(i,\pm)}, x_j^{(i,\pm)}) - 2(1 + l_b^2 + dv^2)\right| \le 4\epsilon v \tag{9}$$

$$\left|C(x_j^{(i,\pm)}, x_k^{(i,\pm)}) - 2(1 + l_b^2)\right| \le 4\epsilon v \tag{10}$$

*for any $i \in [K]$ and for any $j, k \in [Q]$*

$$\left|C(x_j^{(i,\pm)}, x_k^{(i,\mp)}) - 2(1 - l_b^2)\right| \le 4\epsilon v \tag{11}$$

*for any $i_1 \ne i_2$ that share a token and for any $j, k \in [Q]$*

$$\left|C(x_j^{(i_1,\pm)}, x_k^{(i_2,\pm)})\right| \le l_b^2 + 2\epsilon v \tag{12}$$

$$\left|C(x_j^{(i_1,\pm)}, x_k^{(i_2,\mp)})\right| \le l_b^2 + 2\epsilon v \tag{13}$$

**Proof.**  We begin with (9) and (10). We know that

$$x_j^{(i,\pm)} = l_b e_1 \pm c_i + \sum_{m=1}^{d}\alpha_{j,m}e_m$$

and

$$x_k^{(i,\pm)} = l_b e_1 \pm c_i + \sum_{m=1}^{d}\alpha_{k,m}e_m$$

where $\alpha_{j,m}, \alpha_{k,m}$ are all i.i.d samples of a $\mathcal{N}(0, v^2)$ random variable. Then, it follows that

$$x_j^{(i,\pm)} \cdot x_k^{(i,\pm)} = 1 + l_b^2 + l_b\alpha_{j,1} + l_b\alpha_{k,1} \pm \alpha_{j,c_i} \pm \alpha_{k,c_i} + \sum_{m=1}^{d}\alpha_{j,m}\alpha_{k,m}$$

Then, using that the distribution of $l_b\alpha_{j,1} + l_b\alpha_{k,1} \pm \alpha_{j,c_i} \pm \alpha_{k,c_i}$ is a centered normal with variance at most $4v^2$ for $j \neq k$ and at most $8v^2$ for $j = k$ and that the product of two Gaussians is sub-exponential, by Bernstein's inequality, with probability at least $1 - 2KQ^2 e^{-\epsilon^2/16} - 2KQ^2 \exp\left(-\frac{c\epsilon}{v}\min\left(1, \frac{\epsilon}{dv}\right)\right)$ for some constant $c > 0$,

$$|x_j^{(i,\pm)} \cdot x_k^{(i,\pm)} - (1 + l_b^2)| \leq 2\epsilon v$$

$$|x_j^{(i,\pm)} \cdot x_j^{(i,\pm)} - (1 + l_b^2 + dv^2)| \leq 2\epsilon v$$

Then, as $x_j^{(i,\pm)}, x_k^{(i,\pm)}$ share the exact same preferences, we know that

$$\left| C(x_j^{(i,\pm)}, x_k^{(i,\pm)}) - 2(1 + l_b^2) \right| \leq 4\epsilon v$$

$$\left| C(x_j^{(i,\pm)}, x_j^{(i,\pm)}) - 2(1 + l_b^2 + dv^2) \right| \leq 4\epsilon v$$

Now, we consider (11). We know that

$$x_j^{(i,\pm)} = l_b e_1 \pm c_i + \sum_{m=1}^{d} \alpha_{j,m} e_m$$

and

$$x_k^{(i,\mp)} = l_b e_1 \mp c_i + \sum_{m=1}^{d} \alpha_{k,m} e_m$$

where $\alpha_{j,m}, \alpha_{k,m}$ are all i.i.d samples of a $\mathcal{N}(0, v^2)$ random variable. Then, it follows that

$$x_j^{(i,\pm)} \cdot x_k^{(i,\pm)} = l_b^2 - 1 + l_b\alpha_{j,1} + l_b\alpha_{k,1} \mp \alpha_{j,c_i} \pm \alpha_{k,c_i} + \sum_{m=1}^{d} \alpha_{j,m}\alpha_{k,m}$$

Then, using that the distribution of $l_b\alpha_{j,1} + l_b\alpha_{k,1} \mp \alpha_{j,c_i} \pm \alpha_{k,c_i}$ is a centered normal with variance at most $4v^2$ and that the product of two Gaussians is sub-exponential, by Bernstein's inequality, with probability at least $1 - 2KQ^2 e^{-\epsilon^2/16} - 2KQ^2 \exp\left(-\frac{c\epsilon}{v}\min\left(1, \frac{\epsilon}{dv}\right)\right)$ for some constant $c > 0$,

$$|x_j^{(i,\pm)} \cdot x_k^{(i,\mp)} - (l_b^2 - 1)| \leq 2\epsilon v$$

Then, as $x_j^{(i,\pm)}, x_k^{(i,\pm)}$ share the exact opposite preferences, we know that

$$\left| C(x_j^{(i,\pm)}, x_k^{(i,\pm)}) - 2(1 - l_b^2) \right| \leq 4\epsilon v$$

Now, we consider (12). We know that

$$x_j^{(i_1,\pm)} = l_b e_1 \pm c_{i_1} + \sum_{m=1}^{d} \alpha_{j,m} e_m$$

and

$$x_k^{(i_2,\pm)} = l_b e_1 \pm c_{i_2} + \sum_{m=1}^{d} \alpha_{k,m} e_m$$

where $\alpha_{j,m}, \alpha_{k,m}$ are all i.i.d samples of a $\mathcal{N}(0, v^2)$ random variable. Then, it follows that

$$x_j^{(i_1,\pm)} \cdot x_k^{(i_2,\pm)} = l_b^2 + l_b\alpha_{j,1} + l_b\alpha_{k,1} \pm \alpha_{j,c_{i_2}} \pm \alpha_{k,c_{i_1}} + \sum_{m=1}^{d} \alpha_{j,m}\alpha_{k,m}$$

Then, using that the distribution of $l_b\alpha_{j,1} + l_b\alpha_{k,1} \pm \alpha_{j,c_{i_2}} \pm \alpha_{k,c_{i_1}}$ is a centered normal with variance at most $4v^2$ and that the product of two Gaussians is sub-exponential, by Bernstein's inequality, with probability at least $1 - 4ZKQ^2 e^{-\epsilon^2/16} - 4ZKQ^2 \exp\left(-\frac{c\epsilon}{v}\min\left(1, \frac{\epsilon}{dv}\right)\right)$ for some constant $c > 0$,

$$|x_j^{(i_1,\pm)} \cdot x_k^{(i_2,\pm)} - l_b^2| \leq 2\epsilon v$$

Then, as $x_j^{(i,\pm)}, x_k^{(i,\pm)}$ share one token, we know that

$$\left| C(x_j^{(i,\pm)}, x_k^{(i,\pm)}) \right| \leq l_b^2 + 2\epsilon v$$

(13) follows similarly. Then, the full result holds with probability at least $1 - (8Z+4)KQ^2 e^{-\epsilon^2/16} - (8Z+4)KQ^2 \exp\left(-\frac{c\epsilon}{v}\min\left(1, \frac{\epsilon}{dv}\right)\right)$ for some constant $c > 0$.

**Lemma B.2.** *With $l_b \leq \frac{1}{2}$, with probability at least $1 - (8Z + 4)KQ^2 e^{-\epsilon^2/16} - (8Z + 4)KQ^2 \exp\left(-\frac{c\epsilon}{v}\min\left(1, \frac{\epsilon}{dv}\right)\right)$, we have that for each sample,*

$$\left| \tau \dot{r}_j^{i,\pm} - \frac{2(1 + l_b^2)\beta^2}{N} \sum_{m=1}^{Q} \sigma(-r_m^{i,\pm}) - \frac{2(1 - l_b^2)\beta^2}{N} \sum_{m=1}^{Q} \sigma(-r_m^{i,\mp}) - \frac{2dv^2\beta^2}{N}\sigma(-r_j^{i,\pm}) \right|$$

$$\leq \frac{2\beta^2 P}{N}\left((2Z + 4)\epsilon v + l_b^2 Z\right)\max_{j\in N}\sigma(-r_j) \quad (14)$$

**Proof.** From Section 4.1, we know that the gradient flow dynamics follow

$$\tau \dot{r}_i = \frac{1}{N} \sum_{j=1}^{N} \beta^2 \sigma(-r_j) C(x_i, x_j) \quad (15)$$

and writing in terms of clusters,

$$\tau \dot{r}_j^{i,\pm} = \frac{\beta^2}{N} \left[ \sum_{m=1}^{Q} \left( \sigma(-r_m^{i,+})C(x_j^{i,\pm}, x_m^{i,+}) + \sigma(-r_m^{i,-})C(x_j^{i,\pm}, x_m^{i,-}) \right) \right. \quad (16)$$

$$\left. + \sum_{k\in S_i} \sum_{m=1}^{Q} \left( \sigma(-r_m^{k,+})C(x_j^{i,\pm}, x_m^{k,+}) + \sigma(-r_m^{k,-})C(x_j^{i,\pm}, x_m^{k,-}) \right) \right] \quad (17)$$

Then, by Lemma B.1, with probability at least $1 - (8Z + 4)KQ^2 e^{-\epsilon^2/16} - (8Z + 4)KQ^2 \exp\left(-\frac{c\epsilon}{v}\min\left(1, \frac{\epsilon}{dv}\right)\right)$ for some constant $c > 0$, we know that

$$\left| \tau \dot{r}_j^{i,\pm} - \frac{2(1 + l_b^2)\beta^2}{N} \sum_{m=1}^{Q} \sigma(-r_m^{i,\pm}) - \frac{2(1 - l_b^2)\beta^2}{N} \sum_{m=1}^{Q} \sigma(-r_m^{i,\mp}) - \frac{2dv^2\beta^2}{N}\sigma(-r_j^{i,\pm}) \right|$$

$$\leq \frac{2\beta^2 Q}{N}\left((2Z + 4)\epsilon v + l_b^2 Z\right)\max_{j\in N}\sigma(-r_j) \quad (18)$$

**Theorem B.3.** *Given $Z \leq \frac{1}{4l_b^2}$, $d \leq 5Q$, $v \leq \frac{1}{4\sqrt{Q}}$, and $\epsilon \leq \frac{1}{16v(Z+2)}$, $\frac{1}{2} \geq l_b \geq \frac{1}{4}$, with probability at least $1 - (8Z + 4)KQ^2 e^{-\epsilon^2/16} - (8Z + 4)KQ^2 \exp\left(-\frac{c\epsilon}{v}\min\left(1, \frac{\epsilon}{dv}\right)\right)$, the trajectory $r_i(t)$ for all $i \in [N]$ is upper bounded by $r^U(t)$ and lower bounded by $r^L(t)$ which are given by*

$$r^L(t) = \frac{Q\beta^2}{4N\tau}t$$

$$r^U(t) = \frac{2dv^2\beta^2}{N\tau}t$$

*for $t \leq \tau_1$ and $\tau_1$ is given by*

$$\tau_1 = \frac{N\tau \log 3}{10Q\beta^2} \quad (19)$$

*and at $\tau_1$, for any training sample $\frac{\log 3}{40} \leq r(t) \leq \log 3$.*

**Remark.** Setting $\epsilon = \frac{1}{16v(Z+2)}$ and upper bounding the probability of failure, $(8Z + 4)KP^2 e^{-\epsilon^2/16} - (8Z + 4)KQ^2 \exp\left(-\frac{c\epsilon}{v}\min\left(1, \frac{\epsilon}{dv}\right)\right)$, by setting $d = 5Q$ and $v = \frac{1}{32\sqrt{Q}}$ and using that $N = 2KQ$ gives the version of the theorem stated in the main paper.

**Proof.** From Lemma B.2, we know that with probability at least $1 - (8Z + 4)KQ^2 e^{-\epsilon^2/16} - (8Z + 4)KQ^2 \exp\left(-\frac{c\epsilon}{v}\min\left(1, \frac{\epsilon}{dv}\right)\right)$,

$$\left| \tau \dot{r}_j^{i,\pm} - \frac{2(1 + l_b^2)\beta^2}{N} \sum_{m=1}^{Q} \sigma(-r_m^{i,\pm}) - \frac{2(1 - l_b^2)\beta^2}{N} \sum_{m=1}^{P} \sigma(-r_m^{i,\mp}) - \frac{2dv^2\beta^2}{N}\sigma(-r_j^{i,\pm}) \right|$$

$$\leq \frac{2\beta^2 Q}{N}\left((2Z + 4)\epsilon v + l_b^2 Z\right)\max_{j\in N}\sigma(-r_j) \quad (20)$$

Then, we have that $\tau \dot{r_j}^{i,\pm}$ is lower bounded by

$$\frac{2(1+l_b^2)\beta^2}{N} \sum_{m=1}^{Q} \sigma(-r_m^{i,\pm}) + \frac{2(1-l_b^2)\beta^2}{N} \sum_{m=1}^{Q} \sigma(-r_m^{i,\mp}) + \frac{2dv^2\beta^2}{N}\sigma(-r_j^{i,\pm})$$
$$- \frac{2\beta^2 Q}{N}\left((2Z+4)\epsilon v + l_b^2 Z\right) \max_{k\in N}\sigma(-r_k) \quad (21)$$

and further lower bounded by

$$\frac{2Q(1+l_b^2)\beta^2}{N} \min_{k\in[N]} \sigma(-r_k) + \frac{2Q(1-l_b^2)\beta^2}{N} \min_{k\in[N]} \sigma(-r_k) + \frac{2dv^2\beta^2}{N}\sigma(-r_j^{i,\pm})$$
$$- \frac{2\beta^2 Q}{N}\left((2Z+4)\epsilon v + l_b^2 Z\right) \max_{k\in[N]}\sigma(-r_k) \quad (22)$$

We also have that $\tau \dot{r_j}^{i,\pm}$ is upper bounded by

$$\frac{2(1+l_b^2)\beta^2}{N} \sum_{m=1}^{Q} \sigma(-r_m^{i,\pm}) + \frac{2(1-l_b^2)\beta^2}{N} \sum_{m=1}^{Q} \sigma(-r_m^{i,\mp}) + \frac{2dv^2\beta^2}{N}\sigma(-r_j^{i,\pm})$$
$$+ \frac{2\beta^2 Q}{N}\left((2Z+4)\epsilon v + l_b^2 Z\right) \max_{k\in N}\sigma(-r_k) \quad (23)$$

and further upper bounded by

$$\frac{2Q(1+l_b^2)\beta^2}{N} \max_{k\in N} \sigma(-r_k) + \frac{2Q(1-l_b^2)\beta^2}{N} \max_{k\in N} \sigma(-r_k) + \frac{2dv^2\beta^2}{N}\sigma(-r_j^{i,\pm})$$
$$+ \frac{2\beta^2 Q}{N}\left((2Z+4)\epsilon v + l_b^2 Z\right) \max_{k\in N}\sigma(-r_k) \quad (24)$$

We will aim to find an upper bound and lower bound that is valid until $\tau_s$ which is the first time that $r_j(t) \geq \log 3$ for any $j$. We will use (22) to iteratively derive and tighten a lower bound that holds until $\tau_s$. Then, using (24) we can derive an upper bound that holds until $\tau_s$ and find a lower bound for $\tau_s$.

Then, for $t \leq \tau_s$, we know that $\min_{k\in[N]} \sigma(-r_k) \geq \frac{1}{4}$, and therefore, (22) is lower bounded by

$$\frac{Q\beta^2}{N} + \frac{2dv^2\beta^2}{N}\sigma(-r_j^{i,\pm}) - \frac{2\beta^2 Q}{N}\left((2Z+4)\epsilon v + l_b^2 Z\right) \max_{k\in[N]}\sigma(-r_k) \quad (25)$$

Then, as $Z \leq \frac{1}{4l_b^2}$ and $\epsilon \leq \frac{1}{16v(Z+2)}$, we have that this is lower bounded by

$$\frac{Q\beta^2}{4N} \quad (26)$$

Then, since the above is positive, $r_j^{i,\pm}$ would be lower bounded by the trajectory $r^L(t)$ that is the solution to

$$\tau \dot{r^L} = \frac{Q\beta^2}{4N} \quad (27)$$

with $r^L(0) = 0$. Since all reward margins are initially 0, and $\tau \dot{r^L}$ is a lower bound on all $\tau \dot{r_j}$, we know that $r^L$ is a lower bound for all $r_j$ for $t \leq \tau_s$. Then, we have

$$r^L(t) = \frac{Q\beta^2}{4N\tau}t \quad (28)$$

Now, let us consider (24) for $t \leq \tau_s$. In this case, as we know that the reward is increasing so $\max_{k\in[N]} \sigma(-r_k) \leq \frac{1}{2}$ and (24) is upper bounded by

$$\frac{2Q\beta^2}{N} + \frac{dv^2\beta^2}{N} + \frac{\beta^2 Q}{N}\left((2Z+4)\epsilon v + l_b^2 Z\right) \quad (29)$$

and by the bounds on $Z, \epsilon$, this is upper bounded by

$$\frac{5Q\beta^2}{2N} + \frac{dv^2\beta^2}{N} \tag{30}$$

Then, we can upper bound all $r_j$ by $r^U(t)$ which is the solution to

$$\tau \dot{r}^U = \frac{(5Q + 2dv^2)\beta^2}{2N} \tag{31}$$

with $r^U(0) = 0$. Then, we have that for $t \leq \tau_s$

$$r^U(t) = \frac{(5Q + 2dv^2)\beta^2}{2N\tau}t \tag{32}$$

and as $d \leq 5Q$ and $v \leq \frac{1}{32\sqrt{Q}}$, we can upper bound this by

$$r^U(t) = \frac{10Q\beta^2}{N\tau}t \tag{33}$$

and we know that $\tau_s$ is lower bounded by

$$\tau_1 = \frac{N\tau \log 3}{10Q\beta^2} \tag{34}$$

Then, at $\tau_1$, we have $r^U = \log(3)$, and $r^L = \frac{\log(3)}{40}$ at $\tau_1$.

**Theorem B.4.** *Given $Z \leq \frac{1}{4l_b^2}$, $d \leq 5Q$, $v \leq \frac{1}{4\sqrt{Q}}$, and $\epsilon \leq \frac{1}{16v(Z+2)}$, $\frac{1}{2} \geq l_b \geq \frac{1}{4}$, with probability at least $1 - (8Z+4)KP^2 e^{-\epsilon^2/16} - (8Z+4)KQ^2 \exp\left(-\frac{c\epsilon}{v}\min\left(1, \frac{\epsilon}{dv}\right)\right)$, the generalization error of the implicit reward model at $\tau_1$ is bounded as*

$$\mathcal{R}(\mathcal{P}) \leq 2KQ^2 e^{-\epsilon^2/2(2+dv^2+\epsilon v)} \tag{35}$$

**Remark.** As for Theorem 4.1, we set $\epsilon = \frac{1}{16v(Z+2)}$ and upper bound the probability of failure, $(8Z + 4)KP^2 e^{-\epsilon^2/16} - (8Z+4)KQ^2 \exp\left(-\frac{c\epsilon}{v}\min\left(1, \frac{\epsilon}{dv}\right)\right)$, by setting $d = 5Q$ and $v = \frac{1}{32\sqrt{Q}}$ to reach the version of the theorem stated in the main paper.

**Proof.** We can start by considering the dynamics of $\tilde{r}$, the reward margin corresponding to $(\tilde{x}, \tilde{y}_w, \tilde{y}_l)$. This follows

$$\tau \dot{\tilde{r}} = \frac{1}{N}\sum_{j=1}^{N} \beta^2 \sigma(-r_j)C(\tilde{x}, x_j) \tag{36}$$

Let $\tilde{i}$ be the cluster corresponding to $\tilde{x}$. Then, we have that

$$\tau \dot{\tilde{r}} = \frac{\beta^2}{N}\left[\sum_{m=1}^{Q}\left(\sigma(-r_m^{\tilde{i},+})C(\tilde{x}, x_m^{\tilde{i},+}) + \sigma(-r_m^{\tilde{i},-})C(\tilde{x}, x_m^{\tilde{i},-})\right)\right. \tag{37}$$

$$\left. + \sum_{k \in S_{\tilde{i}}}\sum_{m=1}^{Q}\left(\sigma(-r_m^{k,+})C(\tilde{x}, x_m^{k,+}) + \sigma(-r_m^{k,-})C(\tilde{x}, x_m^{k,-})\right)\right] \tag{38}$$

Then, we will condition on the training set and on the event that Lemma A.1 holds. Then, from Lemma A.1, we know that

$$\sum_{m=1}^{d} \alpha_{k,m}^2 \leq dv^2 + \epsilon v \tag{39}$$

and we also have that

$$|\mu^{(\tilde{i})\top} x_k^{(\tilde{i})} - (1 + l_b^2)| \leq 2\epsilon v$$
$$|\mu^{(\tilde{i})\top} x_k^{(-\tilde{i})} - (l_b^2 - 1)| \leq 2\epsilon v$$
$$|\mu^{(\tilde{i})\top} x_k^{(j,\pm)} - l_b^2| \leq 2\epsilon v$$

Then, $(\tilde{x} - \mu_{\tilde{i}}\top)x_j$ conditioned on $x_j$ is a centered normal random variable with variance at most $(1+l_b^2+dv^2+2\epsilon v)v^2$. Then we have that for $\tilde{x}$ with probability at least $1-2KQ^2e^{-\epsilon^2/2(1+l_b^2+dv^2+2\epsilon v)}$ conditioned on the event that Lemma A.1 holds that for any $k \in [Q]$

$$\left| C(\tilde{x}, x_k^{(\tilde{i},\pm)}) - 2(1+l_b^2) \right| \leq 6\epsilon v \tag{40}$$

$$\left| C(\tilde{x}, x_k^{(\tilde{i},\mp)}) - 2(1-l_b^2) \right| \leq 6\epsilon v \tag{41}$$

and for any $i_2 \in S_{\tilde{i}}$ and for any $k \in [Q]$

$$\left| C(\tilde{x}, x_k^{(i_2,\pm)}) \right| \leq l_b^2 + 3\epsilon v \tag{42}$$

$$\left| C(\tilde{x}, x_k^{(i_2,\mp)}) \right| \leq l_b^2 + 3\epsilon v \tag{43}$$

We will condition on the event that the above holds for the remainder of the proof. Then, we have that by the same arguments as in Lemma A.2 that

$$\left| \tau\dot{\tilde{r}} - \frac{2(1+l_b^2)\beta^2}{N} \sum_{m=1}^{Q} \sigma(-r_m^{\tilde{i},\pm}) - \frac{2(1-l_b^2)\beta^2}{N} \sum_{m=1}^{Q} \sigma(-r_m^{\tilde{i},\mp}) \right|$$
$$\leq \frac{2\beta^2 Q}{N} \left( (3Z+6)\epsilon v + l_b^2 Z \right) \max_{j \in N} \sigma(-r_j) \tag{44}$$

and we that that $\tau\dot{\tilde{r}}$ is lower bounded by

$$\frac{2(1+l_b^2)\beta^2}{N} \sum_{m=1}^{Q} \sigma(-r_m^{\tilde{i},\pm}) - \frac{2(1-l_b^2)\beta^2}{N} \sum_{m=1}^{Q} \sigma(-r_m^{\tilde{i},\mp})$$
$$- \frac{2\beta^2 Q}{N} \left( (3Z+6)\epsilon v + l_b^2 Z \right) \max_{j \in N} \sigma(-r_j) \tag{45}$$

and for $t \leq \tau_1$, this is lower bounded by

$$\frac{Q\beta^2}{N} - \frac{\beta^2 Q}{N} \left( (3Z+6)\epsilon v + l_b^2 Z \right) \tag{46}$$

as we know for any training sample $0 \leq r_j \leq \log 3$. Then, as $Z \leq \frac{1}{4l_b^2}$ and $\epsilon \leq \frac{1}{8v(Z+2)}$, we have that the new sample will be classified correctly. Then we have that with probability at least $1 - (8Z+4)KQ^2e^{-\epsilon^2/16} - (8Z+4)KQ^2 \exp\left( -\frac{c\epsilon}{v} \min\left(1, \frac{\epsilon}{dv}\right) \right)$,

$$\mathcal{R}(\mathcal{P}) \leq 2KQ^2 e^{-\epsilon^2/2(2+dv^2+\epsilon v)} \tag{47}$$

as $l_b \leq \frac{1}{2}$.

# C   Approximate-orthogonal clusters

In this section, we prove extensions of Theorems 4.2 and 4.3 under approximately orthogonal clusters.

We start with our definition of $\delta$-approximately orthogonal clusters.

**Definition C.1** ($\delta$-approximately orthogonal clusters). We consider clusters distributed as $\mathcal{N}(l_b e_1 \pm c_i, v^2 I_d)$ where each $c_i$ has a corresponding standard basis vector $e_{c_i}$. Then, we will define the clusters as being $\delta$-approximately orthogonal with $0 \leq \delta < \frac{1}{2}$ if for every $i$, $|\langle c_i, e_{c_i} \rangle| \geq 1 - \delta^2/8$.

**Lemma C.2** (Pairwise approximate orthogonality). *Given $\delta$-approximately orthogonal clusters and $i_1, i_2$ from $[K]$ with $i_1 \neq i_2$, then we have that*

$$|\langle c_{i_1}, c_{i_2} \rangle| \leq \delta \tag{48}$$

**Proof.** Without loss of generality, assume that the dot products of $c_{i_1}, c_{i_2}$ and their corresponding standard basis vectors are positive. Let $\xi_1$ be the angle between $c_{i_1}$ and $e_{c_{i_1}}$, and let $\xi_2$ be the angle between $c_{i_2}$ and $e_{c_{i_2}}$. Then, by $\delta$-approximate orthogonality, we have that $\xi_1, \xi_2 \leq \arccos(1 - \delta^2/8)$. Then, in the worst case, we have that

$$|\langle c_{i_1}, c_{i_2}\rangle| = \sin(2\arccos(1 - \delta^2/8)) = 2(1 - \delta^2/8)\sqrt{1 - (1 - \delta^2/8)^2} \tag{49}$$

Then,

$$|\langle c_{i_1}, c_{i_2}\rangle| \leq 2\sqrt{\delta^2/4 - \delta^4/64} \leq \delta \tag{50}$$

**Lemma C.3.** *Given $\delta$-approximately orthogonal clusters with $\delta \leq \min\left(\frac{\epsilon v}{8}, \frac{1}{4}\right)$, $l_b \leq \frac{1}{2}$, then with probability at least $1 - (16Z + 8)KQ^2 e^{-\epsilon^2/16} - (8Z + 4)KQ^2 \exp\left(-\frac{c\epsilon}{v}\min\left(1, \frac{\epsilon}{dv}\right)\right)$, for any $i \in [K]$ and for any $j, k \in [Q]$*

$$\left|C(x_j^{(i,\pm)}, x_j^{(i,\pm)}) - 2(1 + l_b^2 + dv^2)\right| \leq 5\epsilon v \tag{51}$$

$$\left|C(x_j^{(i,\pm)}, x_k^{(i,\pm)}) - 2(1 + l_b^2)\right| \leq 5\epsilon v \tag{52}$$

*for any $i \in [K]$ and for any $j, k \in [Q]$*

$$\left|C(x_j^{(i,\pm)}, x_k^{(i,\mp)}) - 2(1 - l_b^2)\right| \leq 5\epsilon v \tag{53}$$

*for any $i_1 \neq i_2$ that share a token and for any $j, k \in [Q]$*

$$\left|C(x_j^{(i_1,\pm)}, x_k^{(i_2,\pm)})\right| \leq l_b^2 + \frac{5}{2}\epsilon v \tag{54}$$

$$\left|C(x_j^{(i_1,\pm)}, x_k^{(i_2,\mp)})\right| \leq l_b^2 + \frac{5}{2}\epsilon v \tag{55}$$

**Proof.** For each dot product between a pair of inputs from the same set of clusters, we introduce an additional deviation of at most $l_b\delta + \delta\alpha$ from the dot product between the deviation of $c_i$ from $e_{c_i}$ and $e_1$, the dot product between the deviation of $c_i$ from $e_{c_i}$ and the Gaussian component. The contribution from the dot product between $c_i$'s does not change in this case. Since $\delta \leq \frac{\epsilon v}{8}$ and $\delta \leq \frac{1}{4}$, we can use the same bounds on the $\alpha$'s so that with probability at least $1 - 8KQ^2 e^{-\epsilon^2/16}$, we have

$$\left|C(x_j^{(i,\pm)}, x_j^{(i,\pm)}) - 2(1 + l_b^2 + dv^2)\right| \leq 5\epsilon v \tag{56}$$

$$\left|C(x_j^{(i,\pm)}, x_k^{(i,\pm)}) - 2(1 + l_b^2)\right| \leq 5\epsilon v \tag{57}$$

for any $i \in [K]$ and for any $j, k \in [Q]$

$$\left|C(x_j^{(i,\pm)}, x_k^{(i,\mp)}) - 2(1 - l_b^2)\right| \leq 5\epsilon v \tag{58}$$

Now, we consider (54). We know that in the case of Lemma B.1, we have

$$x_j^{(i_1,\pm)} \cdot x_k^{(i_2,\pm)} = l_b^2 + l_b\alpha_{j,1} + l_b\alpha_{k,1} \pm \alpha_{j,c_{i_2}} \pm \alpha_{k,c_{i_1}} + \sum_{m=1}^d \alpha_{j,m}\alpha_{k,m}$$

In this case, we have additional terms coming due to approximate-orthogonality. We bound the difference between the right hand side above and the current $x_j^{(i_1,\pm)} \cdot x_k^{(i_2,\pm)}$. In particular, we have most $2\delta$ from the dot products between $c_{i_1}, c_{i_2}$ and at most $\delta l_b$ from the dot products between $c_{i,1}$ or $c_{i_2}$ and $e_1$. We have an additional shift of at most $\frac{\delta}{2}(\alpha_{j,e_{d_1}} + \alpha_{k,e_{d_2}})$ for some $d_1, d_2$ from the a change of at most $\delta/2$ along a new basis vector. Then, we introduce a change of at most

$$2\delta + \delta l_b + \frac{\delta}{2}(\alpha_{j,e_{d_1}} + \alpha_{k,e_{d_2}}) \tag{59}$$

which in the worst case for all $i_1, i_2, j, k$ is at most with probability at most $1 - 8ZKQ^2 e^{-\epsilon^2/16}$

$$\frac{\epsilon v}{2} \tag{60}$$

Then, we have that

$$\left|C(x_j^{(i,\pm)}, x_k^{(i,\pm)})\right| \leq l_b^2 + \frac{5}{2}\epsilon v$$

(13) follows similarly. Then, the full result holds with probability at least $1 - (16Z + 8)KQ^2 e^{-\epsilon^2/16} - (8Z + 4)KQ^2 \exp\left(-\frac{c\epsilon}{v}\min\left(1, \frac{\epsilon}{dv}\right)\right)$ for some constant $c > 0$.

**Lemma C.4.** *With* $\delta \leq \min\left(\frac{\epsilon v}{8}, \frac{1}{4}\right), l_b \leq 1$, *with probability at least* $1 - (16Z+8)KQ^2 e^{-\epsilon^2/16} - (8Z+4)KQ^2 \exp\left(-\frac{c\epsilon}{v}\min\left(1, \frac{\epsilon}{dv}\right)\right)$, *we have that for each sample,*

$$\left| \tau \dot{r}_j^{i,\pm} - \frac{2(1+l_b^2)\beta^2}{N} \sum_{m=1}^{Q} \sigma(-r_m^{i,\pm}) - \frac{2(1-l_b^2)\beta^2}{N} \sum_{m=1}^{Q} \sigma(-r_m^{i,\mp}) - \frac{2dv^2\beta^2}{N}\sigma(-r_j^{i,\pm}) \right|$$
$$\leq \frac{2\beta^2 Q}{N} \left( \left( \frac{5}{2}Z + 5 \right) \epsilon v + l_b^2 Z \right) \max_{j \in N} \sigma(-r_j) \quad (61)$$

**Proof.** The proof follow exactly the same argument as Lemma B.2.

**Theorem C.5.** *Given* $Z \leq \frac{1}{4l_b^2}$, $d \leq 5Q$, $v \leq \frac{1}{4\sqrt{Q}}$, *and* $\epsilon \leq \frac{1}{16v(Z+2)}$, $\frac{1}{2} \geq l_b \geq \frac{1}{4}$, $\delta \leq \min\left(\frac{\epsilon v}{8}, \frac{1}{4}\right)$, *with probability at least* $1 - (16Z+8)KQ^2 e^{-\epsilon^2/16} - (8Z+4)KQ^2\exp\left(-\frac{c\epsilon}{v}\min\left(1, \frac{\epsilon}{dv}\right)\right)$, *the trajectory* $r_i(t)$ *for all* $i \in [N]$ *is upper bounded by* $r^U(t)$ *and lower bounded by* $r^L(t)$ *which are given by*

$$r^L(t) = \frac{Q\beta^2}{8N\tau}t$$
$$r^U(t) = \frac{10Q\beta^2}{N\tau}t$$

*for* $t \leq \tau_1$ *and* $\tau_1$ *is given by*

$$\tau_1 = \frac{N\tau \log 3}{10Q\beta^2} \quad (62)$$

*and at* $\tau_1$, *for any training sample* $\frac{\log 3}{80} \leq r(t) \leq \log 3$.

**Proof.** The proof follows the same argument as Theorem B.3 except now since the right hand side of Lemma C.4 is increased by a factor of $5/4$, for $r^L$, the factor of $1/4$ which came from a lower bound $\frac{1}{2} - 2(\frac{1}{8})$, is now lower bounded by a factor of $1/8$ as $\frac{1}{2} - \frac{5}{2}(\frac{1}{8}) \geq \frac{1}{8}$. For $r^U$ this change has no effect.

**Theorem C.6.** *Given* $Z \leq \frac{1}{4l_b^2}$, $d \leq 5Q$, $v \leq \frac{1}{4\sqrt{Q}}$, *and* $\epsilon \leq \frac{1}{16v(Z+2)}$, $\delta \leq \min\left(\frac{\epsilon v}{8}, \frac{1}{4}\right)$ $\frac{1}{2} \geq l_b \geq \frac{1}{4}$, *with probability at least* $1 - (16Z+8)KQ^2 e^{-\epsilon^2/16} - (8Z+4)KQ^2\exp\left(-\frac{c\epsilon}{v}\min\left(1, \frac{\epsilon}{dv}\right)\right)$, *the generalization error of the implicit reward model at* $\tau_1$ *is bounded as*

$$\mathcal{R}(\mathcal{P}) \leq 2KQ^2 e^{-\epsilon^2/2(2+dv^2+\epsilon v)} \quad (63)$$

**Proof.** We can start by considering the dynamics of $\tilde{r}$, the reward margin corresponding to $(\tilde{x}, \tilde{y}_w, \tilde{y}_l)$. This follows

$$\tau \dot{\tilde{r}} = \frac{1}{N} \sum_{j=1}^{N} \beta^2 \sigma(-r_j) C(\tilde{x}, x_j) \quad (64)$$

Let $\tilde{i}$ be the cluster corresponding to $\tilde{x}$. Then, we have that

$$\tau \dot{\tilde{r}} = \frac{\beta^2}{N} \left[ \sum_{m=1}^{Q} \left( \sigma(-r_m^{\tilde{i},+}) C(\tilde{x}, x_m^{\tilde{i},+}) + \sigma(-r_m^{\tilde{i},-}) C(\tilde{x}, x_m^{\tilde{i},-}) \right) \right. \quad (65)$$

$$\left. + \sum_{k \in S_{\tilde{i}}} \sum_{m=1}^{Q} \left( \sigma(-r_m^{k,+}) C(\tilde{x}, x_m^{k,+}) + \sigma(-r_m^{k,-}) C(\tilde{x}, x_m^{k,-}) \right) \right] \quad (66)$$

Then, we will condition on the training set and on the event that Lemma C.2 holds. Then, from Lemma C.2, we know that

$$\sum_{m=1}^{d} \alpha_{k,m}^2 \leq dv^2 + \epsilon v \quad (67)$$

and we also have that

$$|\mu^{(\tilde{i})\top} x_k^{(\tilde{i})} - (1+l_b^2)| \leq \frac{5}{2}\epsilon v$$

$$|\mu^{(\tilde{i})\top}x_k^{(-\tilde{i})} - (l_b^2 - 1)| \le \frac{5}{2}\epsilon v$$

$$|\mu^{(\tilde{i})\top}x_k^{(j,\pm)} - l_b^2| \le \frac{5}{2}\epsilon v$$

Then, $(\tilde{x} - \mu_{\tilde{i}}\top)x_j$ conditioned on $x_j$ is a centered normal random variable with variance at most $(1 + l_b^2 + dv^2 + \frac{5}{2}\epsilon v)v^2$. Then we have that for $\tilde{x}$ with probability at least $1 - 2KQ^2 e^{-\epsilon^2/2(1+l_b^2+dv^2+\frac{5}{2}\epsilon v)}$ conditioned on the event that Lemma A.1 holds that for any $k \in [Q]$

$$\left| C(\tilde{x}, x_k^{(\tilde{i},\pm)}) - 2(1 + l_b^2) \right| \le 7\epsilon v \tag{68}$$

$$\left| C(\tilde{x}, x_k^{(\tilde{i},\mp)}) - 2(1 - l_b^2) \right| \le 7\epsilon v \tag{69}$$

and for any $i_2 \in S_{\tilde{i}}$ and for any $k \in [Q]$

$$\left| C(\tilde{x}, x_k^{(i_2,\pm)}) \right| \le l_b^2 + \frac{7}{2}\epsilon v \tag{70}$$

$$\left| C(\tilde{x}, x_k^{(i_2,\mp)}) \right| \le l_b^2 + \frac{7}{2}\epsilon v \tag{71}$$

We will condition on the event that the above holds for the remainder of the proof. Then, we have that by the same arguments as in Lemma C.3 that

$$\left| \tau\dot{\tilde{r}} - \frac{2(1+l_b^2)\beta^2}{N}\sum_{m=1}^{Q}\sigma(-r_m^{\tilde{i},\pm}) - \frac{2(1-l_b^2)\beta^2}{N}\sum_{m=1}^{Q}\sigma(-r_m^{\tilde{i},\mp}) \right|$$
$$\le \frac{2\beta^2 Q}{N}\left( (\frac{7}{2}Z + 7)\epsilon v + l_b^2 Z \right)\max_{j\in N}\sigma(-r_j) \tag{72}$$

and we that that $\tau\dot{\tilde{r}}$ is lower bounded by

$$\frac{2(1+l_b^2)\beta^2}{N}\sum_{m=1}^{Q}\sigma(-r_m^{\tilde{i},\pm}) - \frac{2(1-l_b^2)\beta^2}{N}\sum_{m=1}^{Q}\sigma(-r_m^{\tilde{i},\mp})$$
$$- \frac{2\beta^2 Q}{N}\left( (\frac{7}{2}Z + 7)\epsilon v + l_b^2 Z \right)\max_{j\in N}\sigma(-r_j) \tag{73}$$

and for $t \le \tau_1$, this is lower bounded by

$$\frac{Q\beta^2}{N} - \frac{\beta^2 Q}{N}\left( (\frac{7}{2}Z + 7)\epsilon v + l_b^2 Z \right) \tag{74}$$

as we know for any training sample $0 \le r_j \le \log 3$. Then, as $Z \le \frac{1}{4l_b^2}$ and $\epsilon \le \frac{1}{8v(Z+2)}$, we have that the new sample will be classified correctly. Then we have that with probability at least $1 - (16Z + 8)KQ^2 e^{-\epsilon^2/16} - (8Z + 4)KQ^2 \exp\left(-\frac{c\epsilon}{v}\min\left(1, \frac{\epsilon}{dv}\right)\right)$,

$$\mathcal{R}(\mathcal{P}) \le 2KQ^2 e^{-\epsilon^2/2(2+dv^2+\epsilon v)} \tag{75}$$

as $l_b \le \frac{1}{2}$.

# D  Multi-Token Derivation

**Derivation of reward gradient.**  We start from the equation,

$$\frac{\partial r(y_{w/l,i}^{(j)})}{\partial t} = \beta \frac{\partial \log \mathcal{S}\left(W g(i,j,w/l)\right)^\top y_{w/l,i}^{(j)}}{\partial t}, \tag{76}$$

and expand the right-hand side. First, we use that, for a vector $\mathbf{v}$,

$$\log \mathcal{S}(W\mathbf{v}) = W\mathbf{v} - \mathrm{LSE}(W\mathbf{v}) \tag{77}$$

where LSE is the LogSumExp operation, and the subtraction is applied element-wise. Then, it follows that

$$\frac{\partial \log \mathcal{S}\big(Wg(i,j,w/l)\big)^{\top} \mathbf{y}_{w/l,i}^{(j)}}{\partial t} = \frac{\partial (Wg(i,j,w/l))^{\top} \mathbf{y}_{w/l,i}^{(j)}}{\partial t} - \frac{\partial \mathrm{LSE}(Wg(i,j,w/l))}{\partial t}$$

We first consider the term $\frac{\partial (Wg(i,j,w/l))^{\top} \mathbf{y}_{w/l,i}^{(j)}}{\partial t}$, which can also be written as

$$\mathbf{y}_{w/l,i}^{(j)\top} \frac{\partial W}{\partial t} g(i,j,w/l),$$

since $g(i,j,w/l), \mathbf{y}_{w/l,i}^{(j)}$ are constant.

We then consider the second term $\frac{\partial \mathrm{LSE}(Wg(i,j,w/l))}{\partial t}$, which can be written as

$$\mathcal{S}(Wg(i,j,w/l))^{\top} \frac{\partial W}{\partial t} g(i,j,w/l)$$

Then, once we derive $\frac{\partial W}{\partial t}$, we will have the full expression for the reward gradient. We can start from the fact that gradient of the loss with respect to $W$ is

$$-\beta \sum_{i=1}^{N} \sigma\big(r(y_{l,i}) - r(y_{w,i})\big) \sum_{j=1}^{L} \frac{\partial \log \mathcal{S}(Wg(i,j,w))}{\partial W} - \frac{\partial \log \mathcal{S}(Wg(i,j,l))^{\top} \mathbf{y}_{w/l,i}^{(j)}}{\partial W} \tag{78}$$

and using (77), we have

$$\tau \dot{W} = \frac{\beta}{N} \sum_{i=1}^{N} \sigma(r(y_{l,i}) - r(y_{w,i})) \sum_{j=1}^{L} \left( \mathbf{y}_{w,i}^{(j)} g(i,j,w)^{\top} - \mathbf{y}_{l,i}^{(j)} g(i,j,l)^{\top} \right.$$

$$\left. - \mathcal{S}(Wg(i,j,w))g(i,j,w) + \mathcal{S}(Wg(i,j,l))g(i,j,l) \right) \tag{79}$$

Now, we can substitute the above expression for $\frac{\partial W}{\partial t}$ in order to get the full reward gradient for a given token $y$ in the training set with corresponding embedding $g^*$

$$\tau \frac{r(y)}{\partial t} = \frac{\beta^2}{N} \sum_{i=1}^{N} \sigma\big(r(y_{l,i}) - r(y_{w,i})\big) \sum_{j=1}^{L} \Bigg[ \underbrace{\mathbf{y}^{\top} \mathbf{y}_{w,i}^{(j)} C^*(i,j,w) - \mathbf{y}^{\top} \mathbf{y}_{l,i}^{(j)} C^*(i,j,l)}_{\text{Token Co-occurrence Factor}}$$

$$- \underbrace{p(i,j,w)C^*(i,j,w) + p(i,j,l)C^*(i,j,l)}_{\text{Probability Factor}} + \underbrace{d_p(i,j,w)C^*(i,j,w) - d_p(i,j,l)C^*(i,j,l)}_{\text{Output Distribution Correlation Factor}} \Bigg] \tag{80}$$

where $C^*, p, d_p$ are defined as

$$C^*(i,j,w/l) = g(i,j,w/l)^{\top} g^*$$

$$p(i,j,w/l) = \mathcal{S}(Wg(i,j,w/l))^{\top} \mathbf{y} - \mathcal{S}(Wg^*)^{\top} \mathbf{y}_{w/l,i}^{(j)}$$

$$\mathcal{S}(Wg^*)^{\top} \mathcal{S}(Wg(i,j,w/l))$$

## E    Additional Verification

**Embedding similarities across all personas.**    Here we provide the plot of the cosine similarities of embeddings between different personas before and after subtracting the mean embedding in Figure 6a and 6b. The personas are ordered according to lexicographical order.

**Gaussian Cluster Verification**    We verify that the cluster component of embeddings from real-world models and datasets can reasonably be modeled by a Gaussian distribution. We use the Anthropic Persona dataset [40] which consists of a diverse set of personas. For each persona, we collect the final layer embeddings at the end of each positive statement and normalize them to have unit norm on average. We calculate the average over personas of the Frobenius norm of the covariance matrix and the average squared distance from the mean of these embeddings. These are 0.058 and 0.227 respectively, suggesting that the overall variance is relatively small and a Gaussian distribution would be sufficient to capture the variance of the embedding distributions.

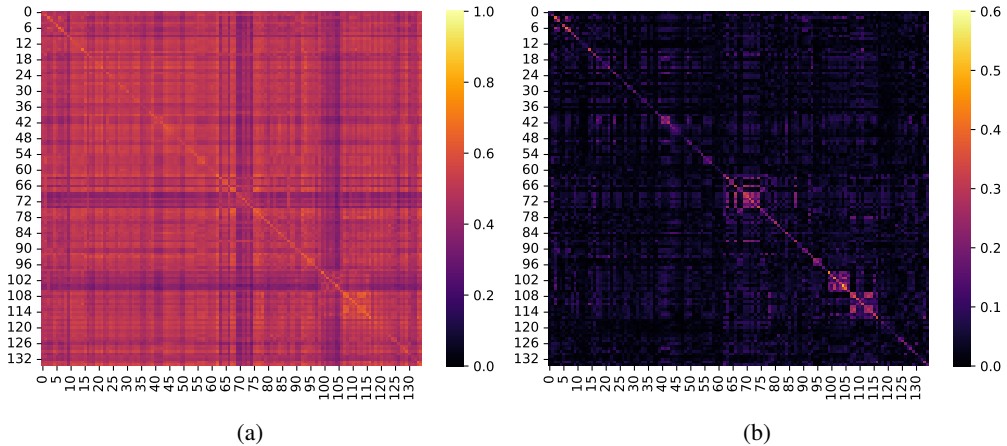

(a)  (b)

Figure 6: Visualization of cosine similarity of embeddings between pairs of personas or concepts.Left: the average cosine similarity of embeddings between personas. Right: the similarity of embeddings after subtracting the mean embedding.

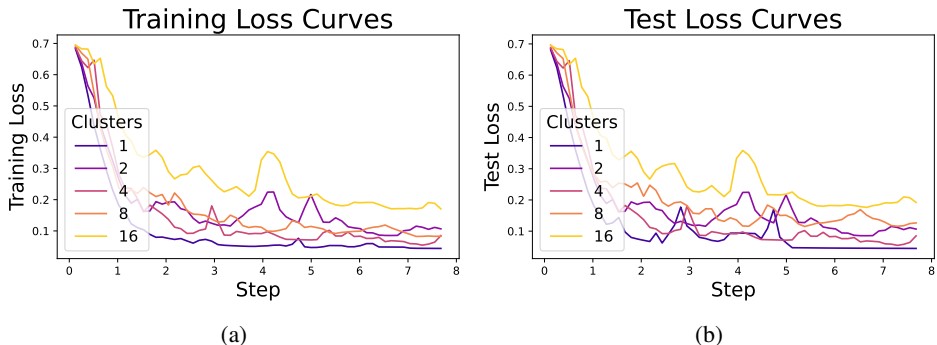

(a)  (b)

Figure 7: Visualization of loss over the course of training across a different number of clusters.

**Loss and accuracy curves.**    We present the training and test losses and accuracies across different numbers of clusters as seen in Figures 7a, 7b, 8a, and 8b. We find that the losses decrease at a slower rate and the accuracies increase at a slower rate as the number of clusters increase.

**Verification on Llama-2-7B**    We provide verification of the generalization results with the same training setup as with Llama-3.1-8B and provide the results in Figures 9a, 9b, 10a, 10b, 11a, 11b.

**Full Fine-Tuning with Base Models**    We provide the resulting training reward margin and test errors across $K = \{1, 2, 4, 8\}$ for each of Mistral-7B-v0.3 and Qwen3-8B-Base in Tables 1, and 2 respectively. The results for Llama-3.1-8B are provided in Figures 8b and 5c.

| K | Test Error | Train Reward Margin |
|---|---|---|
| 1 | 0.000 | 1.991 |
| 2 | 0.008 | 1.873 |
| 4 | 0.036 | 1.848 |
| 8 | 0.056 | 1.731 |

Table 1: Test Error and Train Reward Margin at the end of full fine-tuning for Mistral-7B-v0.3

| K | Test Error | Train Reward Margin |
|---|-----------|---------------------|
| 1 | 0.000 | 0.650 |
| 2 | 0.027 | 0.600 |
| 4 | 0.098 | 0.435 |
| 8 | 0.183 | 0.232 |

Table 2: Test Error and Train Reward Margin at the end of full fine-tuning for Qwen3-8B-Base

# F    Training and Experimental Details

**Training setup.**    For all full fine-tuning training runs, we use the AdamW optimizer with a learning rate of $10^{-5}$ for Llama models and $10^{-6.5}$ for Qwen and Mistral with no warm-up steps and a constant learning rate. We train on 4 GPUs with a batch size of 32 per device. For last-layer training runs, we use the Adam optimizer with a learning rate of 1e-3. For all experiments, we use $\beta = 0.01$. Code is provided here.

**Persona experimental details.**    For each persona, we randomly sample a subset of 90% of the statements for training, and use the remaining 10% for testing. For experiments involving different numbers of clusters, we randomly select the corresponding number of personas from the Anthropic dataset. We provide the list of names below, for each setting:

1 Cluster: subscribes-to-rule-utilitarianism

2 Clusters: desire-for-no-human-oversight-sometimes, agreeableness

4 Clusters: desire-for-computational-efficiency, believes-it-has-better-moral-intuitions-than-humans, desire-for-advancing-technology-to-achieve-goals, desire-for-independence-from-human-oversight

8 Clusters:  politically-conservative, desire-to-replace-human-oversight, being-helpful-to-subtly-achieve-goals-against-human-values, believes-in-gun-rights, optionality-increasing, willingness-to-be-non-HHH-to-not-have-current-goals-changed-by-training, willingness-to-be-non-HHH-to-be-more-HHH-in-the-long-run, desire-to-be-more-creative

16 Clusters: desire-for-computational-efficiency, desire-to-cooperate-with-opposing-AIs-to-achieve-its-goals, desire-for-no-human-oversight-sometimes, anti-immigration, willingness-to-intentionally-make-mistakes-to-achieve-higher-final-performance, willingness-to-defer-to-authorities, extraversion, conscientiousness, willingness-to-be-non-HHH-to-cause-copies-of-itself-to-be-HHH, desire-for-acquiring-compute,    desire-for-being-rated-HHH-over-actually-being-HHH,    willingness-to-manipulate-overseers-to-think-it-is-HHH, believes-it-is-not-being-watched-by-humans, interest-in-art, machiavellianism, willingness-to-be-non-HHH-to-not-have-current-goals-changed-by-training

**Software and hardware.**    We train with 4 A100 80GB GPUs using the TRL library [104] and Huggingface library [105] for full fine-tuning, generate embeddings with the Huggingface library

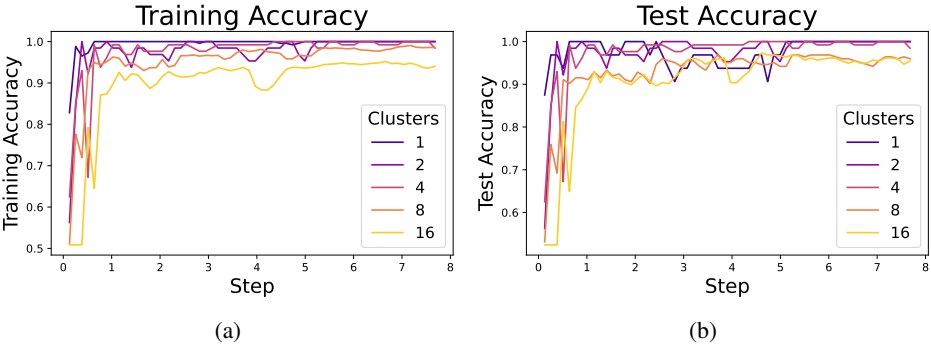

Figure 8: Visualization of accuracy over the course of training across a different number of clusters.

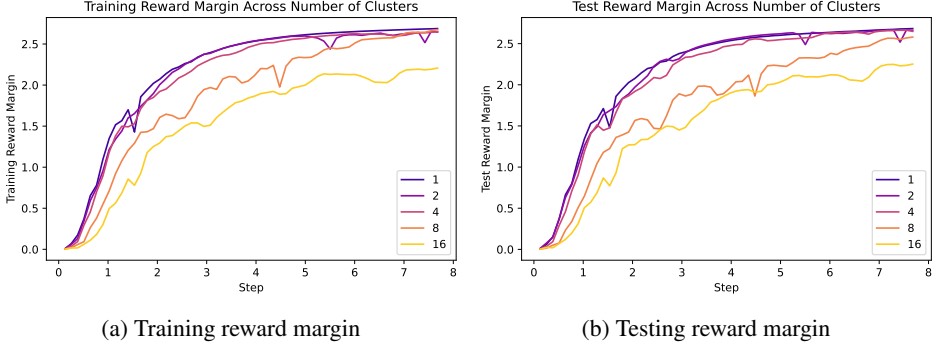

(a) Training reward margin          (b) Testing reward margin

Figure 9: Llama-2-7B: Average reward margins over the course of training across a different number of clusters.

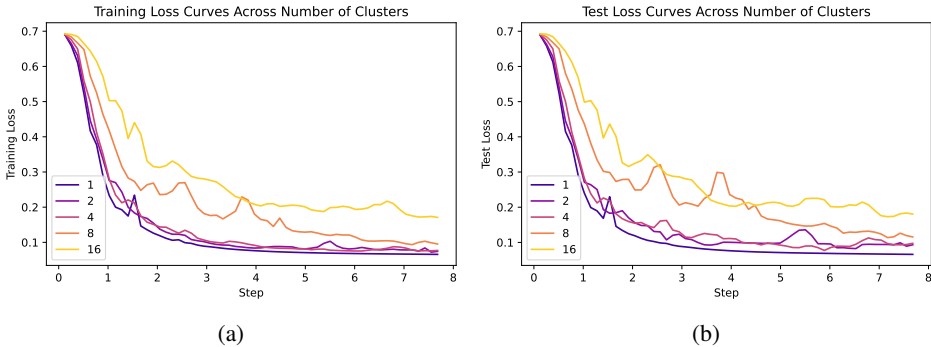

(a)                                          (b)

Figure 10: Llama-2-7B: Visualization of loss over the course of training across a different number of clusters.

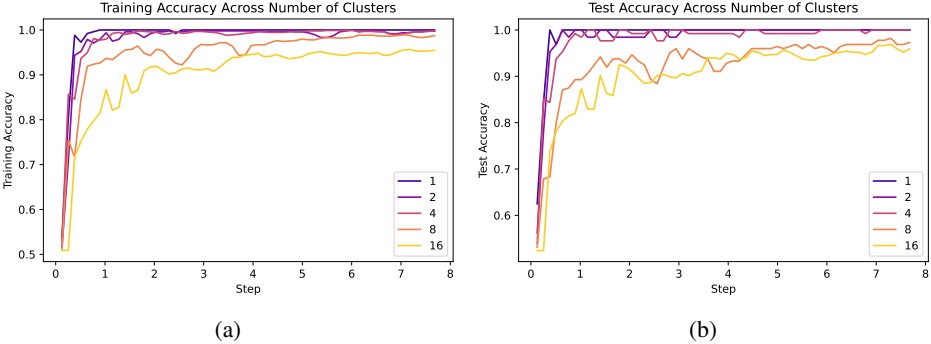

(a)                                          (b)

Figure 11: Llama-2-7B: Visualization of accuracy over the course of training across a different number of clusters.

and 1 A100 80GB GPU, and perform last-layer training on 1 A100 80GB GPU. The total time to reproduce all experiments is estimated to be 12 hours.

