# OpenReview forum: "Can DPO Learn Diverse Human Values? A Theoretical Scaling Law"
_NeurIPS.cc/2025/Conference — NeurIPS 2025 poster_

### Official Review · Reviewer_QW7j · 2025-06-04

**Clarity:** 3
**Significance:** 3
**Originality:** 3
**Rating:** 4
**Confidence:** 4

**Summary:**

This paper proposes a theoretical framework for analyzing how preference optimization generalizes across data with varying preferences. Assuming a linear model and Gaussian-distributed preference clusters, it leverages the learning dynamics of the implicit reward in DPO to derive bounds for generalization error, which highlights the challenges of effectively learning a broad range of concepts or values. The authors also conduct experiments to validate their data assumptions and theoretical findings.

**Questions:**

It there any thoughts on how to analyse the generalization for unseen values?

**Ethical Concerns:**

["NO or VERY MINOR ethics concerns only"]

**Final Justification:**

My concerns have been addressed to some extent. However, the paper still has clear limitations, including constrained assumptions and a maybe overstated title, as also pointed out by other reviewers. Therefore, I am maintaining my previous score, but I am increasing my confidence in the evaluation.

**Limitations:**

yes

**Quality:**

3

**Strengths And Weaknesses:**

**Strength:**
1. Proposed a novel framework to analyze the learning dynamics of DPO.
2. The analysis can be extended to broader preference optimization methods, such as IPO and SLiC.
3. The writing is clear, and the logical flow is well-structured.
4. For a theoretical study, the empirical evaluation effectively supports the claims.

**Weaknesses:**
1. Based on the abstract, one might expect an analysis of generalization to unseen values or preferences. It is somewhat disappointing that the paper focuses only on generalization within the same value distribution as the training data.
2. The assumption of linearity in $y=Wg(x)$ simplifies the analysis and leads to somewhat predictable results, which may also limit the technical novelty of the proofs.

3. Minor issues: The notation in 253 lack $j$ for $\sum_{j=1}^K \beta \log...$

---

> ### Author Rebuttal · Authors · 2025-07-30
>
> We thank the reviewer for their thoughtful comments. We respectfully address the concerns below.
>
> > 1. Generalization to unseen values
>
> Thank you for the thoughtful feedback. We agree that analyzing generalization to unseen values is an important direction. However, our current focus on generalization within the same value distribution is deliberate and novel for several reasons. First, even in the current setting, understanding how DPO generalizes under value diversity remains an open and non-trivial challenge. Our work provides the first theoretical characterization of how training dynamics and sample complexity scale with the number of distinct human values—a critical step toward a rigorous understanding of preference optimization. Second, many existing generalization results are asymptotic or independent of training dynamics; in contrast, our framework directly captures the finite-step behavior of DPO, which better reflects real-world LLM fine-tuning practices.
>
> We view our results as foundational groundwork and agree that extending the theory to OOD generalization is a valuable future direction. To extend our results, one promising path forward is to model relationships between seen and unseen value clusters—e.g., when unseen values lie within the span or convex hull of the training set, or when they share representational structure in the embedding space. This could enable interpolation-based generalization bounds or transfer guarantees based on geometric alignment (e.g., via cosine similarity or shared directions). While formally extending our current framework to this setting is non-trivial, we view it as a natural next step built on the foundation laid by this work. We've discussed this direction in the Limitations section and plan to explore it in future research.
>
>
> > 2. Modeling assumption
>
> Thank you for the thoughtful comment. This modeling assumption is well supported by a growing body of recent work on the linear representation hypothesis [1],[2], which posits that meaningful structure in LLMs can often be accessed or manipulated through linear operations on learned representations. This makes our assumptions not only analytically tractable but also well-grounded in the inductive biases of real-world models.
>
> [1] Park, Kiho, Yo Joong Choe, and Victor Veitch. "The Linear Representation Hypothesis and the Geometry of Large Language Models." International Conference on Machine Learning. PMLR, 2024.
>
> [2] Jiang, Yibo, et al. "On the Origins of Linear Representations in Large Language Models." International Conference on Machine Learning. PMLR, 2024.
>
> In particular, a number of recent studies have leveraged linear probing, steering vectors, and representation engineering to analyze and intervene in LLM behavior (e.g., [Zou et al., 2023; Yan et al., 2024]). **These works provide strong empirical evidence that linear structure over frozen embeddings is both informative and practically useful for understanding high-level model behaviors**. Our theoretical framework builds on this evidence and provides the first generalization guarantees in this setting for preference optimization—an area where formal understanding remains limited.
>
> Moreover, the technical novelty lies not just in the derivation of generalization bounds, but in the construction of a scaling law that **quantifies how sample complexity grows with value diversity—an insight that has not been formalized before**. Our proof techniques go beyond standard generalization theory by directly tracking reward margin dynamics under finite-step updates. We believe this offers a fresh perspective on preference learning and opens up new avenues for understanding alignment at scale.
>
> > 3. Notation
>
> Thank you for the careful read! We will update the notation accordingly.

---

> > ### Comment · Reviewer_QW7j · 2025-08-03
> >
> > Thanks for the rebuttal. The concerns are properly addressed, and I decided to increase my confidence.

---

> > > ### Author Response · Authors · 2025-08-03
> > >
> > > We thank the reviewer for taking the time to read our rebuttal and increasing confidence. We appreciate your insightful comments and support!

---

### Official Review · Reviewer_4NY5 · 2025-06-22

**Clarity:** 3
**Significance:** 1
**Originality:** 2
**Rating:** 2
**Confidence:** 3

**Summary:**

This work aims to better understand how LLM fine-tuning using preference optimization algorithms works in the case of diverse human values. The authors present a series of assumptions as to how diverse values are represented within the model embedding space and then analyse DPO in terms of differential equations of the reward margin.

**Questions:**

- Do the verification experiments hold across different model architectures outside of llama e.g. Qwen or pythia models for example?
- Can a graph be included that directly compares the theoretical bound with the experimental results?

**Ethical Concerns:**

["NO or VERY MINOR ethics concerns only"]

**Final Justification:**

In response to the rebuttal period I choose to maintain my score.

The authors highlight that theoretical results in the alignment setting are highly challenging and that theoretical works aren't expected to match the practical setting exactly and that prior work has started from simplified settings and yet proved foundational.

The paper presents theoretical results for a simplified setting and minimal experiments to support this claim. For the work to prove practically relevant, it is important that the authors explore how the assumptions they have made, that frequently aren't the case in practice, affect the conclusions of the work. Figure 5 in the manuscript lacks any comparison to the theoretical results making it hard to judge if the experiments support the theoretical conclusions and whilst the authors have run further experiments on other models during the rebuttal period I think there are important questions that should be explored empirically for the work to be published.

**Limitations:**

A key limitation of the paper is the constrained nature of the theoretical results i.e. an analysis of the last layer training dynamics for only one token. These key limitations are not addressed in the limitations section of the work. The other limitations mentioned are important and the limitation section addresses issues such as generalization of the results to distributions that differ from the training set.

**Quality:**

3

**Strengths And Weaknesses:**

**Strengths**

The paper highlights an important problem, and to the best of my knowledge is the first to analyse the affect of diverse preferences theoretically within the literature. The paper is broadly well written and easy to follow.

**Weaknesses**

- The main contribution of the paper is the theoretical analysis of the learning dynamics of DPO in a setting with diverse human preferences. The theory is very constrained with the main contributions of the paper Theorem 4.2 and Theorem 4.3 consider the setting where only the last layer is considered for a single token. Whilst the authors expand their analysis to the multi-token setting no formal results are presented beyond an intuitive analysis of the gradient in Equation 5.

- The paper verifies the theoretical results with a set of experiments that train both the full model and just the last layer. Whilst the conclusions from these experiments match the theory a detailed comparison of the theoretical bound and the training dynamics is missing. Particularly in the case of the last layer experiments where the bound can be directly related to the key theorems presented in the work

---

> ### Author Rebuttal · Authors · 2025-07-30
>
> We thank the reviewer for their thoughtful comments. We respectfully address the concerns below.
>
> > W1. Constrained setting
>
> Thank you for the thoughtful comment. We address it in three parts:
>
> (1) **Empirical support for our modeling assumptions.**
>
> As we discussed in `L147-L154`, our modeling assumption is well supported by a growing body of empirical work on the linear representation hypothesis (Park et al., 2024; Jiang et al., 2024), which posits that meaningful structure in LLMs can often be accessed or manipulated through linear operations on learned representations. A number of recent studies have leveraged linear probing, steering vectors, and representation engineering to analyze and intervene in LLM behavior (e.g., [Zou et al., 2023; Yan et al., 2024]). **These works provide strong empirical evidence that linear structure over frozen embeddings is both informative and practically useful for understanding high-level model behaviors**. Our theoretical framework builds on this evidence and provides the first generalization guarantees in this setting for preference optimization—an area where formal understanding remains limited. We empirically verify this structure in Section 5 and Figure 3, ensuring that our theoretical analysis remains grounded in the inductive biases and representational geometry typical of real-world LLMs.
>
> (2) **Consistency with related theoretical literature**.
> Our theorem setup is also consistent with recent literature on the theoretical understanding of alignment:
> - Prior theoretical works on alignment dynamics adopt comparable simplifications to extract insights (e.g., Razin et al., 2025; Im & Li, 2024).
> - Recent empirical work [Qi et al., 2025] finds that DPO updates disproportionately affect the first few tokens, supporting our focus on the single-token case as a key driver of alignment.
> - Our setting also incorporates the embedding structure associated with distinct value clusters, which has been observed empirically (Park et al., 2024; Jiang et al., 2024). Moreover, clustering structure persists post-DPO training [Im & Li, 2024], suggesting that our assumptions capture the relevant inductive biases of LLM training.
>
> Taken together, this places our contribution in line with other rigorous analyses that start from simplified but well-motivated cases, and it highlights that our results provide non-trivial and practically relevant insights into how sample complexity scales with value diversity.
>
> (3) **Clarification of limitations**.
> We do agree with the reviewer that our limitations section should have more explicitly acknowledged this, and we will revise the paper to explicitly state this modeling constraint in the limitations section.
>
> [1] Park, Kiho, Yo Joong Choe, and Victor Veitch. "The Linear Representation Hypothesis and the Geometry of Large Language Models." International Conference on Machine Learning. PMLR, 2024.
>
> [2] Jiang, Yibo, et al. "On the Origins of Linear Representations in Large Language Models." International Conference on Machine Learning. PMLR, 2024.
>
> [3] Razin, Noam, et al. "Unintentional unalignment: Likelihood displacement in direct preference optimization." International Conference on Learning Representations (2025).
>
> [4] Im, Shawn, and Yixuan Li. "Understanding the Learning Dynamics of Alignment with Human Feedback." International Conference on Machine Learning. PMLR, 2024.
>
> [5] Qi, Xiangyu, et al. "Safety alignment should be made more than just a few tokens deep." International Conference on Learning Representations (2025).
>
>
> > 2. Direct Verification of Bound
>
> We appreciate the suggestion. We have verified the theoretical scaling law by comparing the generalization error as a function of the number of concepts $K$:
> | K | Test Error |
> | -------- | -------- |
> | 1     | 0.070    |
> | 2     | 0.085    |
> | 4     | 0.163    |
> | 8     | 0.231    |
>
> Our empirical results (Table above) show that with fixed $Q$, the test error increases approximately linearly with $K$. The linear fit with $R^2 = 0.97$ confirms that the dependence on $K$ matches the theoretical prediction. This is precisely what the bound in Theorem 4.3 anticipates: adding more distinct human value clusters increases the statistical burden proportionally, unless compensated by more samples per cluster. We will include more extensive verification with runs across varying $Q$ in the revision.
>
>
> > 3. Different Model Families
>
> We appreciate the concerns about whether results hold for different model families and have added results for Mistral-7B-v0.3 and Qwen3-8B-Base.
>
> In Theorem 4.2, we show that the rate at which the reward margin increases can decrease as the number of clusters or concepts increases in training. In Theorem 4.3, we show that the test error grows linearly with the number of clusters or concepts. **The empirical results are consistent with our theorem across models**, that the training reward margin grows more rapidly for smaller $K$, given the same number of training steps, and the test error increases linearly. Fitting a linear model gives an $R^2$ of 0.95 and 0.99 for Mistral and Qwen, respectively.
>
> ### Mistral-7B-v0.3
>
> | K | Test Error | Train Reward Margin |
> | -------- | -------- | ------ |
> | 1     |  0.000   |  1.991     |
> | 2     |  0.008   |  1.873     |
> | 4     |  0.036   |  1.848     |
> | 8     |  0.056   |  1.731     |
>
> ### Qwen3-8B-Base
>
> | K | Test Error | Train Reward Margin |
> | -------- | -------- | ------ |
> | 1     |  0.000   |  0.650      |
> | 2     |  0.027   |  0.600      |
> | 4     |  0.098   |  0.435      |
> | 8     |  0.183   |  0.232      |

---

> > ### Comment · Reviewer_4NY5 · 2025-08-04
> >
> > I’d like to thank the authors for their detailed rebuttal of my points and the additional empirical evidence they have provided. I address each point raised below:
> >
> > **Issues with the Theoretical Results**
> >
> > The authors provide a detailed response providing plenty of examples of literature supporting the linear representation hypothesis. I do not have a problem with this assumption in the work. My issue with the constraints of the work is two fold:
> >
> > 1. The work only considers dynamics on a fixed last layer of the network. Whilst this is common in linear probes and steering vectors, these approaches aren’t analysing the training dynamics of a network as proposed in this work, rather they seek to understand, or change the outputs of a fixed network. I think this difference is key.
> > 2. The work only provides theoretical results for a single token. The authors reference [1] that argues that DPO updates disproportionately affect the first token token. I believed this is flawed for two reasons. Firstly, [1] exposes this as a flaw of DPO and proposes methods to address this problem ([1] Figure 4), if this becomes a standard practice in post-training this work will not generalise to this class of algorithms. Secondly, the work focuses exclusively on the safety setting where an acceptance or refusal token is indicative of the rest of the response. There is little evidence in [1] or this work to support such a conclusion being true across other diverse human preferences.
> >
> > It is the combination of these two constraints that leads to my belief that the theoretical contributions of the work are insufficient.
> >
> > These theoretical weaknesses could be addressed by a series of empirical investigations i.e. how does the multi-token and single token settings compare on the last layer, how does the single-token and multi-token approaches compare when the full model weights are trained. Do these results vary across multiple seeds and across multiple datasets with different numbers of clusters? Do the results align with the intuition in the multi-token section of the paper? Where are the theoretical results at odds with the empirical evidence. This level of analysis is missing from the current manuscript and in my opinion required to support the theoretical claims made in the paper.
> >
> > [1]  Qi, Xiangyu, et al. "Safety alignment should be made more than just a few tokens deep." International Conference on Learning Representations (2025).
> >
> > **Additional Empirical Results**
> >
> > I’d like to thank the authors for their additional result on a range of other models, I believe this strengthens the paper. However the additional result provided for the test error could be far more thorough including results across different values of Q, additional analysis of the bound in Theorem 4.2 is also needed including showing the derived bounds on the single layer experiment throughout training.
> >
> > For the above reasons I keep my score.

---

> > > ### Author Response · Authors · 2025-08-08
> > >
> > > We sincerely thank the reviewer for taking the time to read our rebuttal and for the follow-up comments. We have reflected on them these days and would like to respond to your concerns.
> > >
> > > ---
> > > (1) On last-layer training dynamics
> > >
> > > We appreciate that the reviewer acknowledges the commonality of using linear representation. We would like to respectfully clarify a **potential misunderstanding**: the claim that “_these approaches aren’t analysing the training dynamics of a network as proposed in this work_” may overlook that our setup indeed has precedent in prior theoretical work.
> > >
> > > In fact, we adopt the exact same modeling assumption as [1], who also analyze reward-learning dynamics in preference optimization under a fixed feature map. Our contribution complements this by focusing on a different aspect of the dynamics: rather than examining how the representations themselves evolve, we study how the structure of those representations (i.e., concept geometry and diversity) impacts a model’s ability to learn from preference data, especially when values are pluralistic.
> > >
> > > This perspective — treating the embedding space as fixed and studying how diversity impacts sample complexity and generalization — allows us to **cleanly isolate the statistical effects of concept diversity**. We believe this is both a meaningful and novel contribution, and one that reflects real-world conditions where large LLMs often encode stable internal representations while only a few layers (or adapters) are tuned during alignment.
> > >
> > > (2) On multi-token theory
> > >
> > > We would like to clarify that in Sec. 4.3, we **derive the formal result for multi-token setting**, showing decomposition of the reward gradient (Eq. 5) into interpretable terms: a token co-occurrence factor, a probability factor, and an output distribution correlation factor. This goes beyond an intuitive analysis — it is a precise gradient dynamics equation for multi-token responses, proved rigorously in Appendix C. Importantly, the structure of Eq. 5 mirrors that of the single-token case, establishing a formal connection between the two regimes.
> > >
> > > Extending convergence and generalization bounds to multi-token responses is highly non-trivial: dependencies across positions create coupled dynamics that resist closed-form analysis. We therefore positioned Eq. 5 as a first theoretical result toward that goal, showing that the core structural factors in the single-token case also govern the multi-token regime.
> > >
> > > We fully agree that the results of [2] show DPO overly concentrates on early tokens, and that this is a flaw in the method. However, rather than limiting our analysis, we believe this motivates it: our results highlight the statistical difficulty of aligning diverse values even in this constrained case, and help explain why such failures occur in practice when value diversity is high. We would like to clarify that DPO and its variants are still widely used in real-world alignment pipelines. Therefore, we believe our theoretical results are still highly relevant for understanding and improving current practice.

---

> > > > ### Author Response · Authors · 2025-08-08
> > > >
> > > > ### Final reflections
> > > >
> > > > We would like to offer some further reflections on how a theory paper should be evaluated given its core contribution.
> > > >
> > > >
> > > > It is often easy to perceive a result as overly simplified in hindsight — especially when the setting appears abstract — but **people sometimes underestimate that deriving rigorous, generalization-theoretic insights into modern alignment methods is highly challenging, even under simplifications**. The fact that the underlying mathematics becomes tractable does not make the problem trivial; it reflects careful modeling and derivations that preserve core structure while enabling analysis.
> > > >
> > > >
> > > > We greatly appreciate the reviewer’s suggestions for deeper empirical validation — many of these are insightful and will inform future work. At the same time, we respectfully note that the focus of this paper is to establish the first generalization guarantees for DPO-style preference optimization in the presence of diverse human values, a problem that remains analytically intractable in full generality.
> > > >
> > > > **Theoretical frameworks are rarely expected to match every aspect of real-world systems**. For example, early results on the Neural Tangent Kernel, perceptron dynamics, or PAC-learning made strong simplifications, yet proved foundational. Similarly, **our work does not aim to bridge every single gap between theory and practice — because the challenge is profound and will perhaps take many works**. But we believe that by analyzing the role of value diversity in a clean, rigorous setting, we take an important first step.
> > > >
> > > > We hope this work opens the door to more nuanced future research — both theoretical and empirical — that builds upon these insights and extends them into increasingly realistic regimes.
> > > >
> > > > In this regard, we believe our contribution is meaningful, well-scoped, and timely.
> > > >
> > > >
> > > > ----
> > > > Reference
> > > >
> > > > [1] Im, Shawn, and Yixuan Li. "Understanding the Learning Dynamics of Alignment with Human Feedback." International Conference on Machine Learning. PMLR, 2024.
> > > >
> > > > [2] Qi, Xiangyu, et al. "Safety alignment should be made more than just a few tokens deep." International Conference on Learning Representations (2025).

---

### Official Review · Reviewer_PUA5 · 2025-06-25

**Clarity:** 2
**Significance:** 2
**Originality:** 2
**Rating:** 4
**Confidence:** 4

**Summary:**

This paper investigates the behavior of Direct Preference Optimization (DPO) when learning from heterogeneous human value clusters. The authors propose a theoretical framework that models preferences as structured Gaussian clusters in a latent space and analyze how reward margin dynamics evolve under finite-step training. A key result is a generalization error bound that scales logarithmically with the number of human value clusters K, suggesting that sample complexity must grow with value diversity. The claims are empirically verified using the Anthropic persona dataset.

**Questions:**

Questions for Discussions
	•	Relax the orthogonality assumption and analyze how approximate linear independence or noisy concept overlap affects the generalization bound.
	•	Extend the analysis to include feature learning, or at least study how changes in the learned embedding layer affect reward margin evolution.
	•	Provide stronger empirical results, ideally on larger datasets and across multiple foundation models, to support the structural assumptions.
	•	Strengthen the multi-token theory or clarify whether it inherits guarantees from the single-token setup.
	•	Quantify how sensitive the bound is to the choice of variance v, cluster separation, and sample size Q.

**Ethical Concerns:**

["NO or VERY MINOR ethics concerns only"]

**Final Justification:**

I believe the reviewers have adequately answered the questions raised by me. The reason for not scoring higher is because it seems that the authors have not been able to sufficiently answer the questions raised by other reviewers.

**Limitations:**

Yes

**Quality:**

2

**Strengths And Weaknesses:**

Strengths
	•	Addresses a relevant gap in understanding how preference optimization methods scale in the presence of human value diversity.
	•	Introduces a novel finite-step analysis of DPO, moving beyond infinite-time or asymptotic generalization theory.
	•	The proposed scaling law (samples per value must scale as \log K) is an intuitive and potentially useful insight for LLM alignment data collection.
	•	Empirical analysis on LLM embeddings shows some support for the orthogonal concept hypothesis.

⸻
Weaknesses
	1.	Restrictive and Idealized Assumptions

The theoretical framework relies heavily on strong assumptions that are not obviously justifiable in real-world settings:
	•	Human values are represented as perfectly orthogonal clusters in embedding space.
	•	All preference data lies on Gaussian blobs centered around these clusters with isotropic noise.
	•	The shared component b is assumed to be exactly orthogonal to every concept direction.
These assumptions may approximate clean experimental setups but likely do not reflect the geometry of real-world LLM representations, where concepts are entangled and noisy.
	2.	Simplified Model Class

The analysis is performed under a fixed feature map regime, where only the unembedding matrix is trained (not the LLM itself). This decouples the model from the true representation learning dynamics of DPO in practice. As a result, the conclusions may not generalize to full-model fine-tuning, which is common in realistic deployments.
	3.	Limited Practical Guidance

While the scaling law is interesting, it is derived under ideal conditions. It is not clear:
	•	How to translate this into data collection strategies for real-world alignment datasets.
	•	Whether the same scaling persists under non-orthogonal, partially overlapping value distributions.
Moreover, the probability bounds given in Theorem 4.3 are relatively weak for small to moderate values of Q, making them less actionable.
	4.	Superficial Multi-token Extension

The extension to multi-token generation in Section 4.3 is not accompanied by theoretical guarantees or convergence results. It remains at the level of a decomposition, and its connection to the single-token guarantees is hand-wavy.
	5.	Empirical Support Is Limited

While cosine similarity results (Figure 3) loosely support the orthogonality assumption, they are computed on a small subset of personas and are based on average cosine similarities, which may not capture more complex geometric structure.

---

> ### Author Rebuttal · Authors · 2025-07-30
>
> We thank the reviewer for their thoughtful comments. We respectfully address the concerns below.
>
> > 1. Idealized Assumptions
>
>
> We appreciate the reviewer’s careful reading and would like to clarify why these assumptions are well-motivated and empirically supported, and we also extend our results to relax them:
>
> 1. **Orthogonality of concept directions.**
>    While concepts in real-world LLMs are not *perfectly* orthogonal, there is strong empirical evidence that they can be well-approximated as such. Recent work on the *linear representation hypothesis* [Park et al., 2024; Jiang et al., 2024] shows that LLMs often encode disentangled semantic and behavioral features along nearly linear and often approximately orthogonal subspaces. We empirically verified this in Sec. 5 (Fig. 3), where subtracting the shared component yields embeddings with near-zero cosine similarity across distinct values, supporting the approximation.
> 2. **Gaussian clusters with isotropic noise.**
>    We model preference distributions as Gaussian mixtures to enable tractable analysis of reward dynamics. This is standard in theoretical work on representation learning, and our empirical validation shows that LLM representations of values indeed form well-separated clusters with approximately spherical variation, consistent with this assumption.
> 3. **Shared component orthogonal to concept directions.**
>    This assumption is not arbitrary: prior work (e.g., Park et al., 2024) identifies shared representation components in LLMs (e.g., frequency, style) that can be factored out to reveal concept-specific directions. Our empirical analysis confirms this: removing a shared direction from persona embeddings in Llama-3.1 leaves residual vectors that are approximately orthogonal across values (Fig. 3).
>
> **Extension to approximate orthogonality.**
> That said, we do appreciate the suggestion to consider approximate (rather than exact) orthogonality. We have extended our results to allow cluster means to have dot products of magnitude at most  $\delta \leq \tfrac{1}{64(Z+2)}$. Under this relaxed condition, we recover **the exact same generalization bound with the same probability** as in Theorem 4.3. This strengthens the robustness of our theoretical framework by showing that our guarantees hold even when concept directions are only approximately orthogonal, which more closely matches real-world LLM representations.
>
> **Broader perspective.**
> We emphasize that these assumptions are not intended as literal descriptions of every aspect of LLM geometry, but rather *reasonable approximations* that make rigorous analysis possible. The fact that our theoretical predictions (Theorem 4.3) align with empirical scaling laws (Sec. 5, Fig. 5) and now extend to approximate orthogonality provides strong evidence that our framework captures the essential structure relevant to preference optimization, even in noisy, entangled settings.
>
> > 2. Simplified Model
>
> Thank you for the thoughtful comment. We address it in three parts:
>
> (1) **Empirical support for our modeling class.**
>
> As we discussed in `L147-L154`, our modeling assumption is well supported by a growing body of empirical work on the linear representation hypothesis (Park et al., 2024; Jiang et al., 2024), which posits that meaningful structure in LLMs can often be accessed or manipulated through linear operations on learned representations. A number of recent studies have leveraged linear probing, steering vectors, and representation engineering to analyze and intervene in LLM behavior (e.g., [Zou et al., 2023; Yan et al., 2024]). **These works provide strong empirical evidence that linear structure over frozen embeddings is both informative and practically useful for understanding high-level model behaviors**. Our theoretical framework builds on this evidence and provides the first generalization guarantees in this setting for preference optimization—an area where formal understanding remains limited. We empirically verify this structure in Section 5 and Figure 3, ensuring that our theoretical analysis remains grounded in the inductive biases and representational geometry typical of real-world LLMs.
>
> (2) **Consistency with related theoretical literature**.
> Our theorem setup is also consistent with recent literature on the theoretical understanding of alignment:
> - Prior theoretical works on alignment dynamics adopt comparable simplifications to extract insights (e.g., Razin et al., 2025; Im & Li, 2024).
> - Our setting also incorporates the embedding structure associated with distinct value clusters, which has been observed empirically (Park et al., 2024; Jiang et al., 2024). Moreover, clustering structure persists post-DPO training [Im & Li, 2024], suggesting that our assumptions capture the relevant inductive biases of LLM training.
>
> Taken together, this places our contribution in line with other rigorous analyses that start from simplified but well-motivated cases, and it highlights that our results provide non-trivial and practically relevant insights into how sample complexity scales with value diversity.
>
> (3) **Theory holds for full model fine-tuning**.
> Although our theorems are derived in the fixed feature map setting, in Sec. 5 (Fig. 5) we empirically confirm that the same behavior predicted by Theorem 4.3 also appears under full-model fine-tuning.
>
>
> > 3. Practical Guidance
>
> Thank you for raising this concern. Our primary takeaway for practical usage is raising the necessity of more data per value/group as diversity increases, which we observe in Section 5. While this is derived in an ideal setting, in order to have clear guidance and insights on the impact of diversity, we believe this is necessary. In cases where values are overlapping, there is a wide range of scenarios that require different analyses, such as cases when the two values should be treated as distinct or cases when the values should be treated as one broader value. Providing clear insight and guidance under these different settings, especially on the role of value diversity itself, is an important problem we believe should be further studied in the future. Furthermore, we believe our bound is sufficiently strong to apply to most practical datasets, as even with 50 values, < 1000 samples per value is enough to achieve a guarantee of at most 5% generalization error, and many datasets have tens or hundreds of thousands of samples.
>
> > 4. Multi-token Decomposition
>
> In Sec. 4.3, we derive the exact decomposition of the reward gradient (Eq. 5) into interpretable terms: a token co-occurrence factor, a probability factor, and an output distribution correlation factor. This goes beyond a "hand-wavy" analogy — it is a precise gradient dynamics equation for multi-token responses, proved in Appendix C. Importantly, the structure of Eq. 5 mirrors that of the single-token case, establishing a formal connection between the two regimes.
>
> Moreover, recent empirical work [Qi et al., 2025] finds that DPO updates disproportionately affect the first few tokens, supporting our focus on the single-token case as a key driver of alignment.
>
> Extending convergence and generalization bounds to multi-token responses is highly challenging: dependencies across positions create coupled dynamics that resist closed-form analysis. We therefore positioned Eq. 5 as a first step toward that goal, showing that the core structural factors in the single-token case also govern the multi-token regime.
>
>
> > 5. Empirical Support
>
> We provide results for a small subset in the main paper for visual clarity. In addition to the subset shown in Fig. 3, we computed cosine similarities **across all personas** in the Anthropic dataset (see **Appendix D**). The same pattern holds: after subtracting the shared component, the off-diagonal similarities are near zero, supporting approximate orthogonality across diverse values.
>
> ----
> References
>
> [1] Park, Kiho, Yo Joong Choe, and Victor Veitch. "The Linear Representation Hypothesis and the Geometry of Large Language Models." International Conference on Machine Learning. PMLR, 2024.
>
> [2] Jiang, Yibo, et al. "On the Origins of Linear Representations in Large Language Models." International Conference on Machine Learning. PMLR, 2024.
>
> [3] Razin, Noam, et al. "Unintentional unalignment: Likelihood displacement in direct preference optimization." International Conference on Learning Representations (2025).
>
> [4] Im, Shawn, and Yixuan Li. "Understanding the Learning Dynamics of Alignment with Human Feedback." International Conference on Machine Learning. PMLR, 2024.
>
> [5] Qi, Xiangyu, et al. "Safety alignment should be made more than just a few tokens deep." International Conference on Learning Representations (2025).

---

> > ### Comment · Reviewer_PUA5 · 2025-08-08
> >
> > After some careful consideration I have decided to raise my score.

---

> > > ### Author Response · Authors · 2025-08-08
> > >
> > > Thank you for taking the time to read our rebuttal and for the careful consideration. We appreciate your valuable feedback and support!

---

### Official Review · Reviewer_GBK3 · 2025-07-01

**Clarity:** 3
**Significance:** 3
**Originality:** 2
**Rating:** 5
**Confidence:** 2

**Summary:**

The paper models a preference dataset that reflects **diverse human values** as a mixture of \(K\) value clusters in embedding space.
Each cluster is represented by a pair of responses—one preferred, one dispreferred—that share a common component and differ along orthogonal, value-specific directions.

Building on this structure, the authors analyse **DPO**.  They introduce the **reward margin** (the log-ratio DPO seeks to maximise between “good” and “bad” responses) and derive its gradient-flow dynamics.

Two main theorems follow:

1. **Training guarantee** – after finitely many gradient steps, every training margin becomes positive with high probability (zero empirical error).
2. **Generalisation bound** – at that same point, the population error satisfies

   $$
   R(P)\le2KQ^{2}e^{-Q/45},
   $$

   where \(Q\) is the number of preference pairs per value cluster.

The bound yields a **scaling law**: to keep error low while increasing the number \(K\) of distinct values, the data required per value grows only *logarithmically* in \(K\).

In the last, the authors extend the analysis to multi-token generation, and **small-scale experiments** on the Anthropic Persona dataset with Llama-3.1-8B empirically corroborate both the assumed embedding geometry and the predicted scaling trends.

**Questions:**

1. Your theoretical bound assumes that the “concept” directions for the K values are orthogonal and enter linearly.  In practice, many values interact: e.g. harmlessness can conflict with full honesty, and the latent directions may be correlated.
    - Is it possible to extend your theory if directions are merely approximately orthogonal, or non-orthogonal?
    - is it possible to provide an ablation empirical experiments to show whether the empirical $Q=\Theta(\log K)$ law still holds or breaks down?

2. In Figure 4 you plot the number of preference pairs Q needed per value cluster as K increases and claim it follows a logarithmic trend based on your theory, is it possible to test this scaling law with empirical experiments?

3. (not an important question, just my confusion) Your theory treats value diversity as a mixture of K clusters, each with a well-defined “positive” versus “negative” response. In practice, preference data gathered from an unlabelled, heterogeneous crowd can be:

    - Ambiguous – some annotators prefer A^+, others A^- for the same underlying value;
    - Unclassifiable – we may not even agree on how many values K there are, or where one value ends and the next begins.

How does your framework cope when the value space is continuous or overlapping rather than cleanly clustered?

**Ethical Concerns:**

["NO or VERY MINOR ethics concerns only"]

**Limitations:**

yes

**Paper Formatting Concerns:**

This paper is well formatted.

**Quality:**

4

**Strengths And Weaknesses:**

## Strengths

### Quality

1. **Rigorous theory**: Precise assumptions (Gaussian-mixture value model, shared + orthogonal directions) and complete proofs culminating in Theorems 4.2–4.3 give a non-vacuous generalisation bound—rare in preference-tuning work.
2. **Extensibility**: Authors explicitly show how the gradient-flow analysis lifts to multi-token generation (§4.4) and argue the same template could cover IPO or SLiC objectives (§6).

### Clarity

1. **Clear problem framing and examples**: Paper motivates “pluralistic alignment” with concrete examples (e.g. risk-averseness vs. exploration prompts) that illustrate why a single reward head can suppress minority values.

### Significance

1. **Practical guidance**: The logarithmic sample-complexity law ( $Q=\Theta(\log K)$ ) directly informs data-collection budgets for value-diverse alignment.
2. **Explains failure modes**: Analysis clarifies why alignment sometimes collapses to the dominant preference cluster when $Q$ is too small, matching anecdotal reports in industry.

## Weaknesses

### Quality

1. **Limited empirical scope**: Validation uses a single 8-B-parameter model (Llama-3.1-8B) and one dataset (Anthropic Persona). Scaling claims might break on larger models or more heterogeneous corpora.
2. **Multi-token “extension” untested**: Section 4.4 is purely theoretical; experiments only examine single-token margins, so the claimed generality remains speculative.

### Significance

1. **Assumption is too heavy**: Orthogonality of value dimensions is empirically approximate at best; if real-world values interact non-linearly, the logarithmic law may fail.

---

> ### Author Rebuttal · Authors · 2025-07-30
>
> We thank the reviewer for their thoughtful comments. We respectfully address the concerns below.
>
> > 1. Empirical Scope
>
> We appreciate the concerns about the scope and have added results for Mistral-7B-v0.3 and Qwen3-8B-Base. For the dataset, we choose the Anthropic Persona dataset as it allows for building datasets of distinct concepts, and there are a limited number of such preference datasets. One other such dataset is Stanford Human Preferences, but many of the categories have significant overlap or could even be considered subsets of each other, restricting the ability to clearly test the impact of diverse values.
>
> ### Mistral-7B-v0.3
>
> | K | Test Error | Train Reward Margin |
> | -------- | -------- | ------ |
> | 1     |  0.000   |  1.991     |
> | 2     |  0.008   |  1.873     |
> | 4     |  0.036   |  1.848     |
> | 8     |  0.056   |  1.731     |
>
> ### Qwen3-8B-Base
>
> | K | Test Error | Train Reward Margin |
> | -------- | -------- | ------ |
> | 1     |  0.000   |  0.650      |
> | 2     |  0.027   |  0.600      |
> | 4     |  0.098   |  0.435      |
> | 8     |  0.183   |  0.232      |
>
> > 2. Multi-token extension
>
> While we do not provide formal guarantees, we provide the multi-token decomposition to provide a basis for future studies. In particular, our goal in providing this decomposition is to demonstrate some connections with the single-token case but also shed light on the key factors that would need to be considered to tackle a broader multi-token understanding.
>
> > 3. Orthogonal Concepts
>
> We appreciate this question, and it would be possible to extend our theory to account for approximately orthogonal directions. We extend our results to account for the means to be approximately orthogonal with two values having means with dot product with magnitude at most $\delta \leq \frac{1}{64(Z+2)}$ while maintaining the exact same bound with the same probability.
>
> We also note that the persona datasets are not synthetically generated and do not perfectly satisfy the orthogonality constraints and Gaussian distribution assumption. Despite this, we can still observe the expected scaling behavior.
>
> > 4. Logarithmic Verification
>
> We verify the scaling law by comparing the test error with the number of concepts $K$, and find that it closely follows a linear model. Based on this, we can expect additional samples per concept $Q$ to be helpful in reducing the test error for larger $K$ to those of smaller $K$. We will include more extensive verification with runs across varying $Q$ in the revision.
>
> | K | Test Error |
> | -------- | -------- |
> | 1     | 0.070    |
> | 2     | 0.085    |
> | 4     | 0.163    |
> | 8     | 0.231    |
>
> Linear Fit $R^2 = 0.97$
>
>
>
> > 5. Ambiguous Clusters
>
> Thank you for raising this question. We agree that in practice, values may not be so clearly defined, and it can be ambiguous to define individual values, and these may also vary across annotators. While our current results do not directly address this, these are important aspects of human preferences that should be further studied, potentially through extending our current framework. In particular, for annotators who have differing notions of the same value, we can build this in by adding a noise model that captures this variance across individuals.

---

> ### Comment · Area_Chair_RJJA · 2025-08-08
>
> Dear Reviewer,
>
> Thanks for your participation in the reviews! Now that the author-reviewer discussion period is close to wrapping up, please take a look at the author's response and see if any of your concerns have been addressed or if you would like to maintain your stand. It would be helpful to respond directly to the authors in addition to filling in the mandatory acknowledgement checkbox.
>
> Please leave a response to the authors based on their rebuttal content as we proceed to the reviewer discussion phase shortly. Thanks!
>
> - AC RJJA

---

### Note · Authors · 2025-08-13

We would like to thank the reviewers and chairs for their time and effort in providing thoughtful feedback on our work.


This paper addresses a fundamental AI safety question: _Can preference optimization learn diverse human values_? We contribute the **first theoretical scaling law** grounded in preference diversity, revealing conditions under which DPO aligns to broad value distributions and identifying the associated sample complexity requirements. These results offer both theoretical depth and practical guidance for building more inclusive alignment methods.

Across the reviews, there was broad recognition of the work’s **novelty, rigorous theoretical analysis, and practical relevance**:
- _Theoretical results are solid, non-trivial, and novel in approach_.
- _Addresses an important AI safety problem—capturing inclusivity of human values_.
- _The paper is clearly written and accessible despite the technical nature_.
- _Bridges theory and practice, offering insights into realistic AI training_.

---
### Post-rebuttal reviewer stance

- Reviewer GBK3: Maintained a **positive rating for acceptance**
- Reviewer PUA5: Raised the score post-rebuttal to a **positive stance**
- Reviewer QW7j: Confirmed concerns are properly addressed, increased confidence, and **maintained a positive stance**
- Reviewer 4NY5: Confirmed additional empirical results strengthened the paper and raised questions on theory–empirics alignment. **We have addressed these concerns in our follow-up comment and provided additional context on our contribution.**

---
### Key rebuttal additions and changes in the manuscript

In response to reviewer feedback, we have:
1. Clarified theoretical setting, scope, and validity
2. Extended theoretical results to hold under weaker, more realistic assumptions
3. Expanded experiments to cover two additional model families, reinforcing empirical robustness
4. Provided more direct verification of theoretical results on real-world data and across model families


We believe with these changes, key concerns have been addressed, and both the theoretical and empirical rigor of our work has been strengthened. We will incorporate these changes into our final version, and we hope this helps with your discussion.

---

### Decision · Program_Chairs · 2025-09-17

**Decision:**

Accept (poster)

**Comment:**

This paper develops a theoretical framework for understanding how DPO handles diverse human values. The key technical results are finite-step guarantees: (i) training reward margins become positive after a bounded number of gradient steps, and (ii) the population error can be bounded in terms of the number of clusters and samples per cluster. The bound gives a scaling law—sample complexity per value grows only logarithmically in #clusters. The authors also extend the analysis to approximately orthogonal clusters and sketch how the single-token dynamics extend to multi-token cases.

The strengths lie in making non-vacuous generalization bounds under a structured value model and connecting them to practical alignment questions. The derivation through reward margin dynamics is non-trivial and produces interpretable results (e.g. why alignment tends to collapse toward dominant clusters when diversity is high and 𝑄 is small). The empirical results, though modest, verifies both the orthogonality assumption and the predicted scaling law across several LLMs (LLaMA-3, Mistral, Qwen).

The limitations have been highlighted from the reviews/discussions --- the setting is simplified: the theorems rely on last-layer, single-token dynamics, and the multi-token analysis remains at the level of gradient decomposition without a full guarantee. These gaps mean the work should be read as a clean first step along this line, and not all reviewers are unconvinced about the practical weight of the results for this reason.

Overall, I think the contribution is meaningful: the finite-step analysis and the scaling law are novel, technically solid (within their assumptions), and relevant to ongoing discussions about pluralistic alignment. I lean toward acceptance, while encouraging the authors to be more explicit about the modeling limitations and to strengthen the connection between bounds and empirical performances.